# Genetic architecture of spatial electrical biomarkers for cardiac arrhythmia and relationship with cardiovascular disease

The 3-dimensional spatial and 2-dimensional frontal QRS-T angles are measures derived from the vectorcardiogram. They are independent risk predictors for arrhythmia, but the underlying biology is unknown. Using multiancestry genome-wide association studies we identify 61 (58 previously unreported) loci for the spatial QRS-T angle ($N$ = 118,780) and 11 for the frontal QRS-T angle ($N$ = 159,715). Seven out of the 61 spatial QRS-T angle loci have not been reported for other electrocardiographic measures. Enrichments are observed in pathways related to cardiac and vascular development, muscle contraction, and hypertrophy. Pairwise genome-wide association studies with classical ECG traits identify shared genetic influences with PR interval and QRS duration. Phenome-wide scanning indicate associations with atrial fibrillation, atrioventricular block and arterial embolism and genetically determined QRS-T angle measures are associated with fascicular and bundle branch block (and also atrioventricular block for the frontal QRS-T angle). We identify potential biology involved in the QRS-T angle and their genetic relationships with cardiovascular traits and diseases, may inform future research and risk prediction.

Abnormalities of ventricular depolarization and repolarization are a cause of malignant arrhythmia, which are associated with cardiac morbidity and mortality[1]. Mechanisms underlying the relationship of conventional electrocardiographic (ECG) measures with arrhythmogenesis (e.g. the QT interval and QRS duration) have previously been explored and highlight the role of cardiac ion channels. However, the biology reflected by markers derived from the vectorcardiogram is largely unknown[2]. These markers include the spatial (spQRSTa) and frontal (fQRSTa) QRS-T angles, which are the angles between the directions of ventricular depolarization and repolarization in 3- and 2-dimensional space, respectively (Fig. 1)[3]. Previous experimental and theoretical studies have shown that a wider QRS-T angle is determined through local variation in action potential duration and morphology[4,5].

While vectorcardiographic measures are not currently used in routine clinical practice, there has been a resurgence of interest in their potential clinical utility, which has coincided with computational advances for efficient calculation of these markers. Recent studies have reported associations of the spQRSTa and fQRSTa with risk for arrhythmogenesis, sudden cardiac death and cardiac-related mortality[6–8]. In a population-based study, an abnormal spQRSTa was associated with a five-fold increased risk of cardiac and sudden death. No other conventional cardiovascular or ECG measure provided higher hazard ratios[9]. These measures may also be broad markers of cardiovascular risk, and associations have been reported with cardiomyopathies and cardioembolic stroke[10,11]. Improved knowledge of these markers will increase our understanding of these clinical relationships and has potential to identify new biology that is not captured by conventional ECG measures. Genome-wide association studies (GWAS) allow investigation of intermediate phenotypes and complex diseases to identify candidate genes and pathways that contribute to the underlying biology without a predefined hypothesis[12]. A previous GWAS meta-analysis for the spQRSTa ($N$ = 13,826) identified 3 independent loci, with candidate genes involved in cardiac conduction and development[13]. However, this study was limited by a small sample size, and no GWAS has investigated the fQRSTa.

✉ e-mail: tereshl@ccf.org; p.b.munroe@qmul.ac.uk

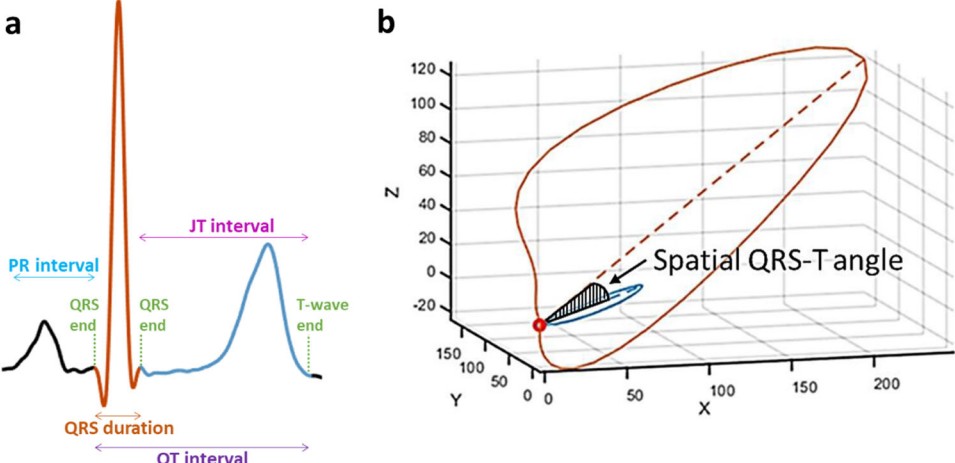

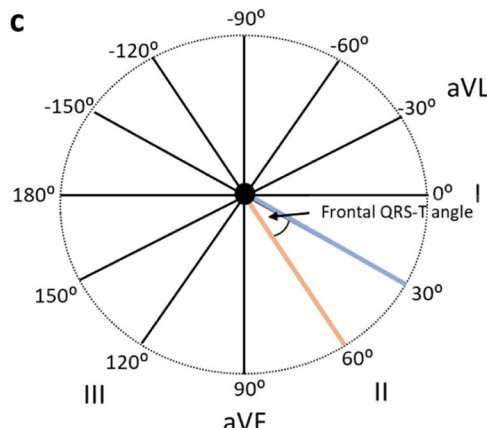

**Fig. 1 | Graphical representation of the spQRSTa and fQRSTa alongside a single electrocardiogram lead signal. a** Single lead electrocardiogram (ECG) signal with classical measures QRS duration and the QT interval labelled. The dark orange (estimates ventricular depolarization time) and blue (ventricular repolarization time) shaded sections of the signal represent the regions used to calculate the QRS and T-wave axes respectively with multiple ECG leads. **b** The spatial QRS-T angle (spQRSTa) mean is the angle between the mean amplitude of QRS and T-wave spatial loops. These spatial loops can be constructed from the resting 12-lead ECG using a standardised transformation, to produce representative X, Y and Z vectors that can be plotted over time. **c** The frontal QRS-T angle (fQRSTa) is the absolute difference between QRS and T-wave axes in the frontal plane only.

We performed the largest multi-ancestry studies to date for the spQRSTa ($N = 118{,}780$) and fQRSTa ($N = 159{,}715$) to identify additional candidate genes and pathways enriched for these markers, to advance our understanding of their genetic relationship with other ECG traits and cardiovascular disease, and to enhance the interpretation of existing and future clinical studies.

## Results

### Meta-analysis of QRS-T angle GWAS
Our primary multi-ancestry GWAS meta-analysis for spQRSTa had a total sample size of 118,780 individuals, including European (81.3%), Hispanic/Latino (10.7%) and African (7%) ancestries from 14 studies. The multi-ancestry GWAS meta-analysis for fQRSTa included 159,715 individuals from 23 studies and a similar ancestral composition (Supplementary Data 1–3, Supplementary Note 1). Ancestry-stratified analyses were also conducted. Due to the non-normal distribution of the traits, results are for the rank-based inverse normal transformed phenotype, with reference to corresponding effect sizes from the raw-phenotype analyses (degrees [°]) for clinical interpretation. No inflation of tests statistics was identified, but early deviation from the reference line was observed in Quantile-Quantile (Q-Q) plots for multi-ancestry and

European-ancestry meta-analyses (driven by a locus on chromosome 17; Supplementary Figs. 1 and 2).

### Genome-wide significant loci
In multi-ancestry meta-analyses, we identified a total of 61 (58 previously unreported) and 11 lead genome-wide significant (GWS; $P < 5 \times 10^{-8}$) variants at independent loci associated with spQRSTa and fQRSTa, respectively (Figs. 2 and 3, Supplementary Data 4 and 5). All lead variants for fQRSTa mapped within a locus reported for spQRSTa. All previously reported loci for spQRSTa (*NFIA*, *HAND1* and *TBX3*) were GWS and were the most significant loci. A total of 51 and 9 GWS independent loci were identified in European ancestry meta-analyses for spQRSTa and fQRSTa, respectively. All loci were also GWS in the corresponding multi-ancestry analysis, except one fQRSTa locus (*TTN*; Supplementary Data 4 and 5).

### Conditional analyses and heritability estimates in European ancestry individuals
To identify additional signals, Genome-wide Complex Trait Analysis (GCTA, v1.26.0)[14] was performed using European ancestry UK Biobank (UKB) participant meta-analysis summary statistics from 33,960 individuals. The analyses identified conditionally independent

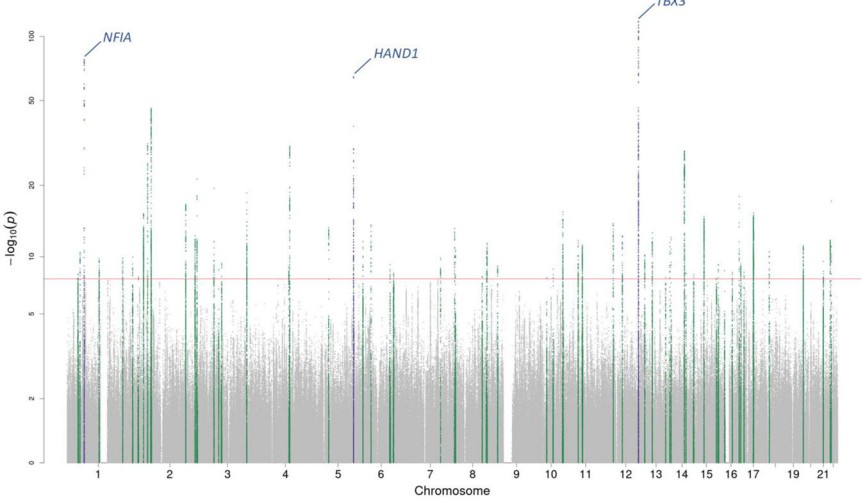

**Fig. 2 | Manhattan plot for the spQRSTa multi-ancestry meta-analysis.** Manhattan plot for the spatial QRS-T angle (spQRSTa) meta-analysis. Two-sided *P*-values are plotted on the -log₁₀ scale (Y-axis). The red horizontal line indicates genome-wide significance ($P < 5 \times 10^{-8}$). Variants within the boundaries of loci previously reported for the spatial QRS-T angle are labelled with the candidate gene and colored blue. Variants at previously unreported loci are green.

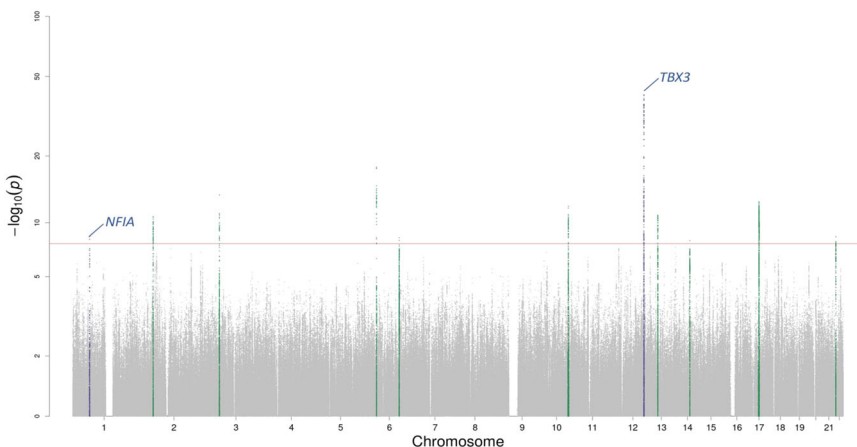

**Fig. 3 | Manhattan plot for the fQRSTa multi-ancestry meta-analysis.** Manhattan plot for the frontal QRS-T angle (fQRSTA) meta-analysis. Two-sided *P*-values are plotted on the -log₁₀ scale (Y-axis). The red horizontal line indicates genome-wide significance ($P < 5 \times 10^{-8}$). Variants within the boundaries of loci previously reported for the spatial QRS-T angle are labelled with the candidate gene and colored blue. Variants at previously unreported loci are green.

variants at 4 loci for spQRSTa and at 2 loci for fQRSTa (Supplementary Data 6).

Common SNP-based heritability was estimated in the same set of UKB participants with BOLT-Restricted Maximum Likelihood (BOLT-REML, v2.3.2) software[15]. Heritabilities of spQRSTa and fQRSTa were 22.3% and 6.8%, respectively (standard error [SE] 1.0%). European ancestry lead and conditionally independent variants explained 4.0% and 0.5% of the variance of spQRSTa and fQRSTa, respectively. Therefore, these variants explain approximately 17.8% and 7.4% of the SNP-based heritability of spQRSTa and fQRSTa, respectively.

**Follow-up of loci for the spatial QRS-T angle**
Over 96% (59/61) of the spQRSTa lead multi-ancestry variants were common (minor allele frequency [MAF] > 0.05). Across all loci, the lead variant with the largest effect size was rs117526881, located upstream of *MYH7* (effect size 3.7° per allele). At each locus, Variant Effect Predictor (VEP, Ensembl release 99) was used to identify potential functional consequences of lead variants and their proxies ($r^2 > 0.8$)[16]. Missense variants were identified at 6 (9.8%) loci (Supplementary Data 7). SIFT or Polyphen-2 prediction tools identified variants that

were likely to be deleterious at 2 loci (*ADPRHL1* and *KANSL1*). The *KANSL1* locus contained missense variants in strong LD with the lead SNP ($r^2 > 0.94$) in multiple genes (*KANSL1, SSPL2C, MAPT* and *LRRC37A2*). The lead variant (or a proxy) of five loci had a Combined Annotation Dependent Depletion (CADD) score ≥20, and were therefore predicted to be among the most deleterious variants in the genome (i.e., in the top 1%; Supplementary Data 8). The low frequency missense variant rs41306688 (effect size −2.5° per allele) at the *ADPRHL1* locus had the highest CADD score (26.7).

To identify variants associated with tissue-specific gene expression in cardiovascular tissues, data were extracted from the Genotype-Tissue Expression (GTEx, v8) project[17]. At 11 loci, the lead variant or a proxy was a significant cis- expression quantitative trait locus (eQTL) variant in cardiac (left ventricular [LV], right atrial appendage [RAA]) or vascular (coronary or aorta artery) tissue (Supplementary Data 9). At 5 loci, we identified support for pairwise colocalization (*BACH* [RAA], *C1QTNF4* [LV, aorta artery], *CDH13* [LV, RAA], *LINC00964* [LV] and *MTSS1* [LV, RAA], and *PKDCC* [LV]; posterior probability [PP] > 0.75).

To predict the effects of gene expression in LV, RAA and vascular tissue on our phenotypes, a transcriptome-wide association study

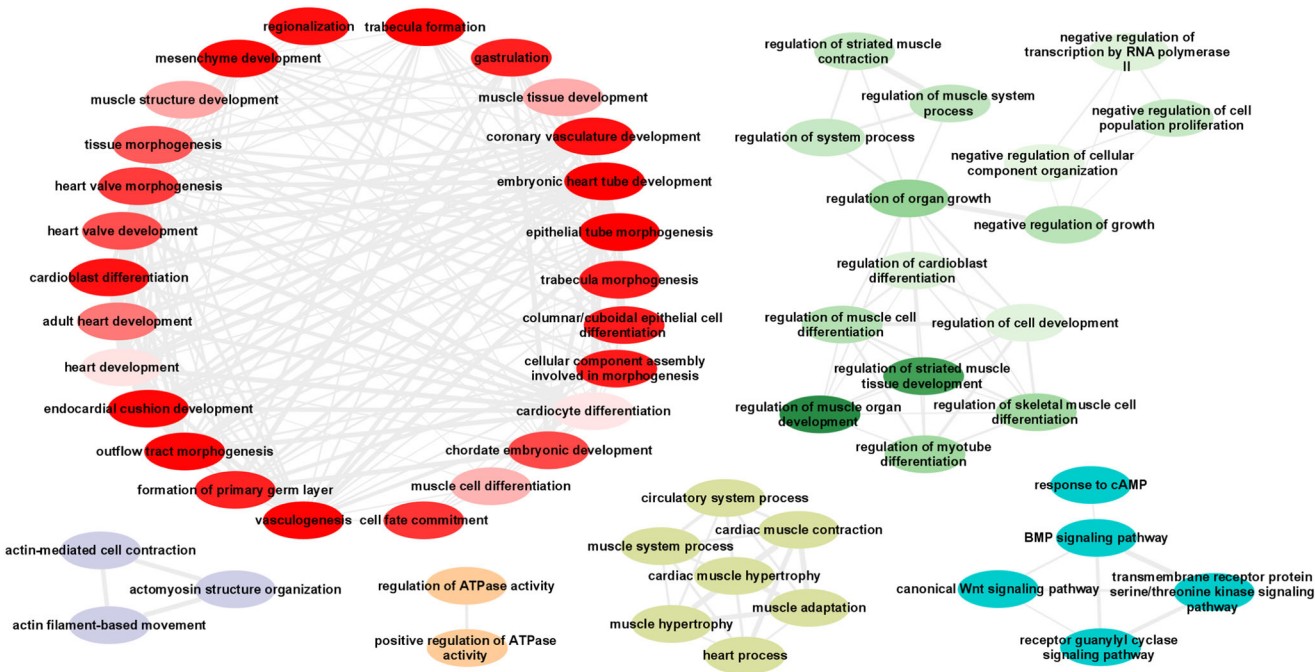

**Fig. 4 | Significant GO biological processes from spQRSTa DEPICT multi-ancestry findings.** All significant (false discovery rate <0.01) multi-ancestry spatial QRS-T angle (spQRSTa) gene-ontology (GO) biological processes from Data-driven Expression-Prioritization Integration for Complex Traits (DEPICT) software were analyzed using the Reduce and Visualize Gene Ontology (REVIGO) web application to remove redundant terms and cluster related nodes. Highly similar GO terms are linked by edges where the line width indicates the degree of similarity. Within each cluster, the colour gradient represents differences in the DEPICT gene-set enrichment two-sided *P*-values, with lighter gradients reflecting smaller enrichment *P*-values (therefore more significant) compared with other nodes in the same cluster.

(TWAS) was performed with S-PrediXcan software. The expression of 33 genes was significantly associated with the spQRSTa (Bonferroni corrected threshold; $P < 3.1 \times 10^{-6}$), 26 of which mapped within GWS loci, and 10 were significant in multiple tissues (Supplementary Data 10). Increased expression was associated with an increase in spQRSTa for 17 genes, whereas an inverse relationship was found for 15 genes (Supplementary Fig. 3). For *TMEM198*, increased expression in the aorta was associated with an increase in spQRSTa, but an inverse relationship was observed in LV tissue. All other genes with significant findings in multiple tissues had concordant directions of effect.

Non-coding variants may influence cardiac electrophysiology through effects on regulatory elements and chromatin folding. We used 40 kb and ~4 kb-resolution long-range chromatin interaction (Hi-C) datasets to identify potential target genes of regulatory variants[18,19]. Promoter interactions were identified at 17 (27.9%) multi-ancestry loci in LV or RV tissues (Supplementary Data 11a, b). GWAS Analysis of Regulatory and Functional Information Enrichment with LD correction (GARFIELD) was used to test for enrichment of variants at DNase 1 hypersensitivity sites in specific tissues using European ancestry summary statistics. The strongest enrichment was in fetal heart tissue ($P < 7.5 \times 10^{-36}$); however, additional tissues were identified, including fetal renal pelvis, adult heart and brain (Supplementary Fig. 4).

With single nucleus Assay for Transposase-Accessible Chromatin using sequencing (snATAC-seq) data, we tested for enrichment of non-coding variants at open chromatin regions, to identify cell-type specific functional effects in adult heart, by utilizing Chromatin Element Enrichment Ranking by Specificity (CHEERS)[20,21]. Significant enrichment was observed across all variants in atrial and ventricular cardiomyocytes (Supplementary Fig. 5).

Reconstituted gene-sets in Data-driven Expression-Prioritisation Integration for Complex Traits (DEPICT) software were used to prioritize potential candidate genes based on overlapping functional pathways[22]. Significant gene-set enrichment (false discovery rate [FDR] < 0.01) was observed in cardiac tissues (ventricle, atrial and atrial appendage) (Supplementary Data 12). Significantly enriched Gene-Ontology (GO) biological processes were extracted from DEPICT pathway analyses (Supplementary Data 13). Redundant GO terms were removed and the remaining processes clustered using the reduce and visualise Gene Ontology (REVIGO) web application[23]. This analysis identified clusters of biological processes involved in: cardiac development (including embryonic heart tube morphogenesis, muscle structure development, trabeculae formation and vasculogenesis); muscle cell differentiation and regulation of organ growth; actin filament-based movement; and cardiac contraction and hypertrophy (Fig. 4). Significant KEGG pathways were dilated, hypertrophic and arrhythmogenic right ventricular cardiomyopathies; cardiac muscle contraction; and arginine and proline metabolism. The top 10 enriched mouse phenotypes included dilated cardiac chambers; ventricular wall thickness (thick and thin); and abnormal cardiac development (Supplementary Data 13).

A summary of bioinformatic annotations for all spQRSTa multi-ancestry loci is provided in Supplementary Data 14. These findings have been supplemented with additional trait-relevant information from: Online Mendelian Inheritance in Man (OMIM)[24]; the International Mouse Phenotyping Consortium[25] (IMP); the Human Protein Atlas[26]; and PubMed literature reviews for each candidate gene. We also performed lookups of each lead variant in the Open Targets Genetics 'Locus to Gene' machine learning gene-prioritization pipeline for further annotations (Supplementary Data 14)[27].

We identified two independent loci in the Hispanic/Latino spQRSTa meta-analysis, including one locus that was not GWS in the multi-ancestry meta-analysis (lead variant rs112628278, multi-ancestry GWAS *P* = 0.01). rs112628278 (nearest gene *VAV2*) is a low frequency Hispanic/Latino variant (MAF = 0.011) and rare among European ancestry individuals (MAF = 0.0002, 1000 Genomes [1000 G] reference panel).

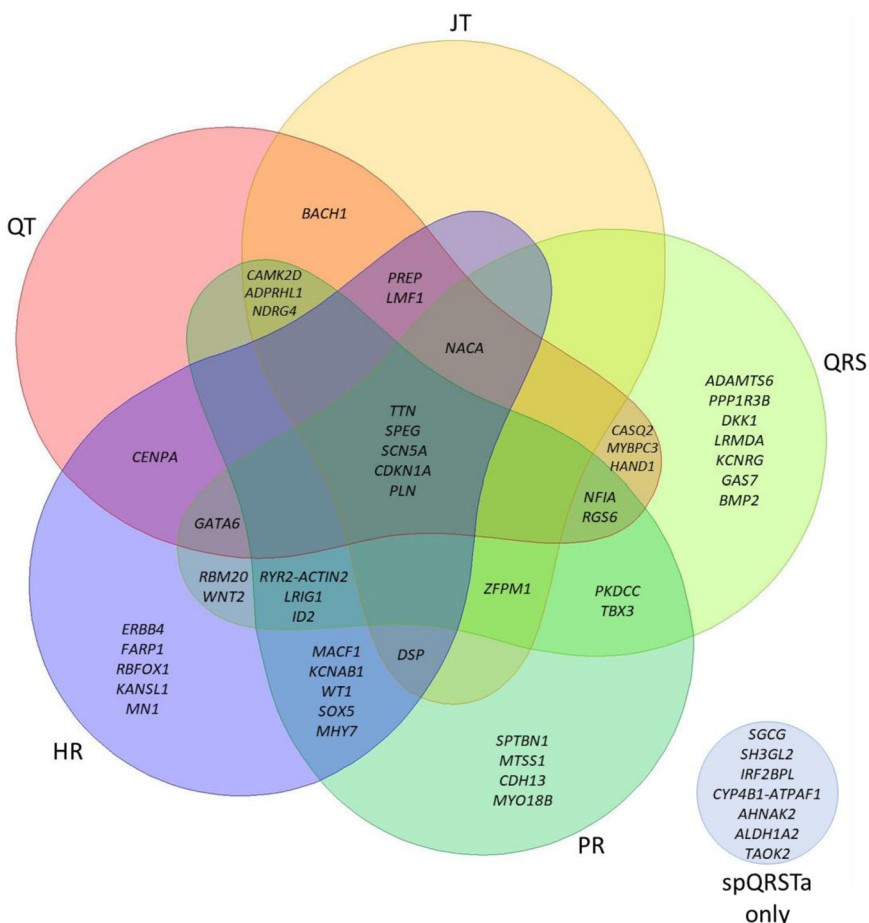

**Fig. 5 | Overlap of multi-ancestry spQRSTa loci with ECG measures.** Venn diagram showing spatial QRS-T angle (spQRSTa) multi-ancestry loci where a lead variant reported for another electrocardiographic ECG measure maps within the locus boundaries. For this figure, ECG measures shown are PR interval (cardiac conduction), QRS duration (ventricular depolarization), QT and JT intervals (ventricular repolarization) and heart rate (HR). Overlap was declared if a lead variant for these ECG measures mapped to within ±500 kb or $r^2 > 0.1$ of a lead variant at a spQRSTa locus. Some loci overlap with other ECG traits (not visualised here but presented in Supplementary Data 15). At seven spQRSTa loci, no overlap was observed with any ECG trait (blue circle bottom right).

One unreported locus (*FAM135B*) identified in the African ancestry spQRSTa meta-analysis showed no evidence for association in the multi-ancestry meta-analysis ($P > 0.05$). The lead variant (rs28377209) has a higher MAF in African ancestry populations, compared with Europeans (0.19 *vs* 0.10).

## Follow-up of loci for the frontal QRS-T angle
Three variants at two loci were significant eQTL variants (LV [*SSXP10, RP11-632C17_A.1*], coronary artery [*GNAZ*]), but there was no support for colocalization (Supplementary Data 9). Eight genes were significant in the TWAS, and overlapped with spQRSTa genes, except for two (*CEP85L* and *MMP11*) (Supplementary Data 10). Tissue-specific promoter interactions were identified for variants at two loci that were not reported for spQRSTa loci (lead variant rs10885011; *FAM124A* and *DLEU7*, rs5030613; *BCR*) (Supplementary Data 11a, b). An unreported locus identified in the African ancestry fQRSTa meta-analysis was not GWS in spQRSTa analyses. The gene nearest to the lead signal is *CCDC60* (Coiled-Coil Domain Containing 60).

## Genetic correlation and overlap of GWS loci with other ECG measures
LD Score Regression (LDSC) software was used to estimate genetic correlations ($r_g$) of spQRSTa and fQRSTa with ECG markers of cardiac conduction (PR interval), ventricular depolarization (QRS duration) and repolarization (QT and JT intervals)[28,29]. There was a high positive genetic correlation between spQRSTa and fQRSTa ($r_g = 0.61$).

Weak positive correlations were observed with PR interval ($r_g = 0.12$, $P = 6 \times 10^{-4}$ for spQRSTa; $r_g = 0.19$, $P = 2.2 \times 10^{-5}$ for fQRSTa). However, no statistically significant correlation was observed with the other ECG traits (Supplementary Fig. 6).

We used additional approaches to interrogate genetic overlaps. First, lead variants reported for other resting ECG traits were extracted and overlap was reported if they mapped within spQRSTa locus boundaries (within $r^2 > 0.1$ or ±500 kb from the lead spQRSTa variant). Despite the low genetic correlations observed genome-wide, 26 (42.6%), 27 (44.3%) and 26 (42.6%) lead multi-ancestry spQRSTa variants mapped to reported PR, QRS and HR loci, respectively (Supplementary Data 15). Fewer variants mapped to reported QT and JT loci (19 [31.1%] and 14 [23%], respectively) (Fig. 5). Of the 7 loci reported for the global electrical heterogeneity trait SAI QRST, 3 lead variants mapped within the boundaries of spQRSTa loci (*SCN5A, MYBPC3* and *NDRG4*).

Next, we performed a pairwise GWAS with GWAS-PW, which uses Bayesian bivariate methods to estimate the probability for each genomic region that a variant affects both traits tested[30]. Across all spQRSTa loci, there was evidence for shared genetic influences at 17 (27.9%), 20 (32.8%), 7 (11.5%), 14 (22.9%) and 12 (19.7%) loci involving PR, QRS, HR, QT and JT, respectively (PP > 0.9). Of the loci that shared effects with QT and JT, 8/14 (57.1%) and 6/12 (50%) loci, respectively, also influenced QRS duration (Supplementary Data 15). The smallest $P$-value for variants at the *NOS1AP* locus in the spQRSTa multi-ancestry meta-analysis was $7.3 \times 10^{-5}$. *NOS1AP* is the locus consistently reported with the strongest QT and JT associations. We performed a sensitivity

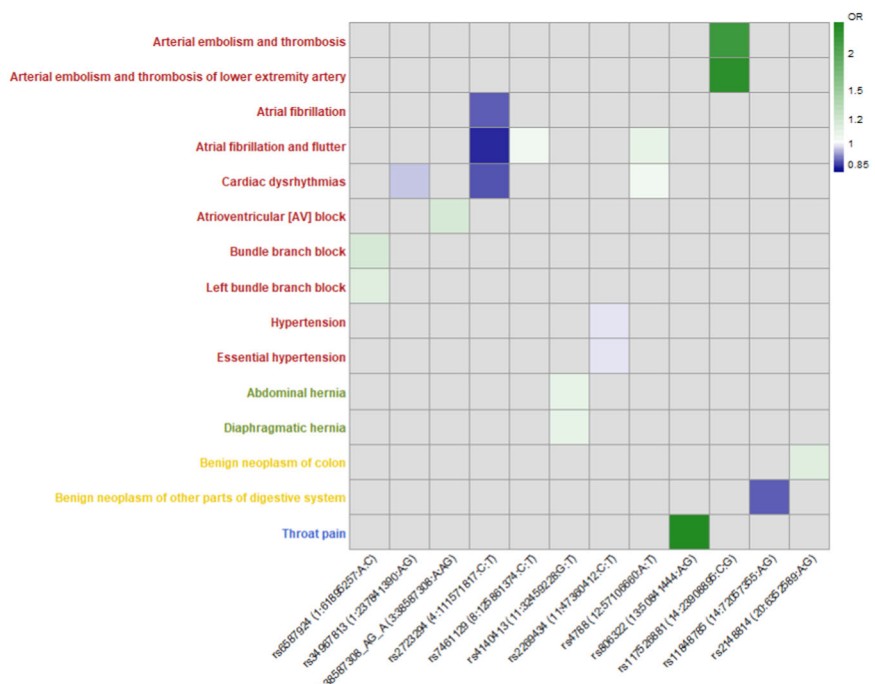

**Fig. 6 | Significant associations observed in phenome-wide association study of lead and conditionally independent spQRSTa variants.** X-axis: Lead variant (RsID [Chromosome: Position (hg19): Allele1: Allele2]) or conditionally independent variant from the spatial QRS-T angle (spQRSTa) European ancestry meta-analysis that had a significant association with a clinical phenotype in UK Biobank. Y-axis: Phenotype derived from hospital episode statistics, with colour coding for each major group (circulatory system; red, digestive system; green, neoplasms; yellow, respiratory; blue). Odds ratios (OR) are color coded according to decreasing (blue) or increasing (green) odds. 3:38587306:A:G was a conditionally independent variant at the *SCN5A* locus.

analysis in ~34,000 UKB individuals to determine whether inclusion of the QT interval as a covariate influenced our findings. Beta estimates and *P*-values were highly correlated (rho[ρ] = 0.99 and 0.96 respectively) across all variants comparing a GWAS with or without the QT interval as a covariate. Also, there was no substantial change in the minimum *P*-value of variants at the *NOS1AP* locus.

At 7 multi-ancestry spQRSTa loci, we observed no overlap with previously reported ECG loci. Candidate genes at these loci include *AHNAK2*, *ALDH1A2*, *SGCG* and *TAOK2*.

## Pleiotropy of genetic variants with other phenotypes

We performed a phenome-wide association study (PheWAS) to identify associations of European ancestry lead and conditionally independent spQRSTa variants with 1301 clinical conditions in 395,758 unrelated individuals European-ancestry individuals. Data on clinical conditions were from hospital episode statistics. Significantly associated conditions included atrial fibrillation, bundle branch block (BBB), atrioventricular block (AVB), arterial embolism and thrombosis, and hypertension (Fig. 6). We also performed lookups of all multi-ancestry lead spQRSTa variants (and proxies) in Phenoscanner (v2), to determine if they appeared in GWAS reports for non-ECG phenotypes and diseases (Supplementary Data 16). Lead variants or proxies at 19 spQRSTa loci (31.1%) had reported associations with blood pressure, anthropometric traits, blood counts, or psychiatric features or disorders ($P < 5 \times 10^{-8}$).

## Association of genetically determined spQRSTa and fQRSTa with cardiovascular disease

Polygenic risk scores (PRSs) were used to explore associations between genetically determined spQRSTa and fQRSTa and relevant cardiovascular diseases. PRSs were calculated by summing the dosage of lead variants from the European-ancestry meta-analysis, weighted by the effect size estimates from the corresponding untransformed analysis. To obtain preliminary β estimates for the association of PRSs

with the directly measured ECG trait, we performed a linear regression adjusting for age, sex, RR interval, BMI, height and 10 genetic principal components, in 33,960 unrelated individuals of European ancestry from UKB. These individuals were included in the GWAS meta-analysis, and therefore β estimates and CIs are biased. However, approximation is useful to aid interpretation of subsequent analyses. Associations observed for each PRS were (β [95% CI]): 5.4° (5.1–5.7) for spQRSTa; and 2.03° (1.8–2.3) for fQRSTa (per standard deviation [SD] increase in the PRS).

Subsequently, each PRS was tested for association with prevalent cases of cardiovascular disease in 395,758 unrelated European ancestry UKB participants who were not in the GWAS meta-analysis (adjusting for sex, age, and 10 genetic principal components). We used a Bonferroni corrected threshold to identify significant findings (0.05/number of conditions tested, $P < 6.3 \times 10^{-3}$). Genetically determined spQRSTa was associated with increased odds for fascicular or bundle branch block (odds ratio [OR] (95% CI) per SD: 1.10 [1.07–1.13]) (Supplementary Fig. 7, Supplementary Data 17). Association of a QRS PRS with fascicular or bundle branch block has been reported[31]. However inclusion of a QRS PRS as a covariate did not substantially change the point estimates (1.09 [1.06–1.13]), supporting an interpretation that the spQRSTa PRS contains independent risk information. There was suggestive evidence for an association with AV block but not at the Bonferroni corrected significance threshold (OR: 1.04 [1.01–1.06], $P = 7.7 \times 10^{-3}$). Genetically determined fQRSTa was significantly associated with fascicular block or bundle branch block (OR 1.05 [1.02–1.08]), and AV block (OR 1.04 [1.01–1.07]).

## No evidence for a causal relationship between spQRSTa and cardiomyopathies

Because candidate genes and pathway analyses indicated potential involvement with cardiomyopathies, we performed Mendelian randomization (MR) studies to test for a causal relationship of genetically determined spQRSTa (as the exposure) with hypertrophic

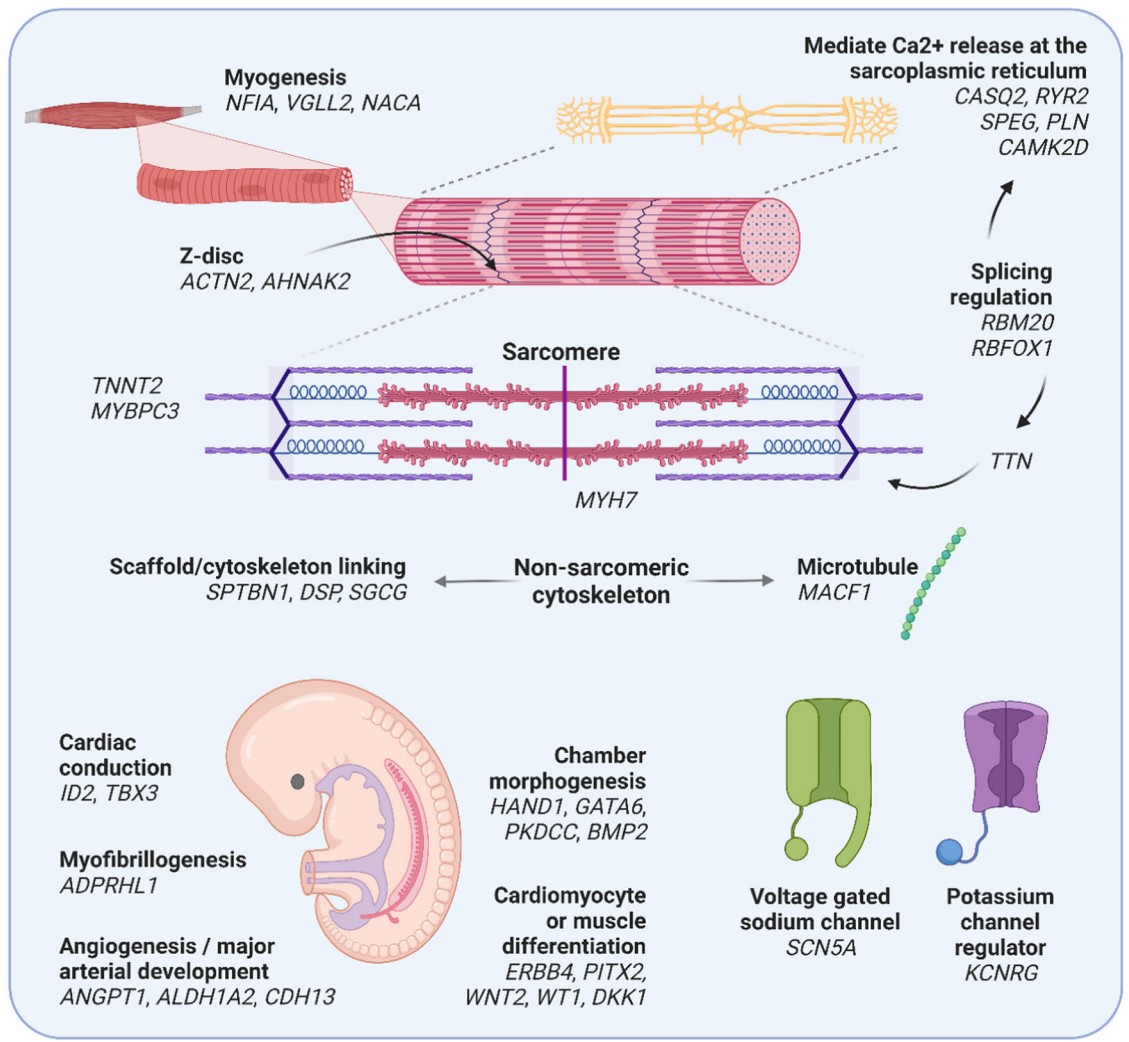

**Fig. 7 | Illustration of candidate genes at spQRSTa multi-ancestry loci and their potential function.** Candidate genes at spatial QRS-T angle (spQRSTa) loci are grouped according to potential roles in embryonic development, cardiac structure and function. *RYR2* and *ACTN2* are candidate genes from the same locus. A summary of the bioinformatic evidence for each gene is presented in Supplementary Data 14. Created using BioRender.com.

cardiomyopathy[32] (HCM) and idiopathic dilated cardiomyopathy[33] (DCM) (as outcomes). Lead variants from multi-ancestry and European spQRSTa meta-analyses were used as instrumental variables (IV). A relationship was suggested with HCM (multi-ancestry: OR 1.01 [1.00–1.02], *P* = 0.004; European: 1.01 [1.00–1.02], *P* = 0.009), using a fixed-effect inverse variance-weighted (IVW) model. However, the association was not supported in sensitivity analyses, including MR-Egger, weighted median and MR-PRESSO analyses (Supplementary Data 18). Similarly, no causal relationship was identified with either sarcomere positive or sarcomere negative HCM cases. There were no significant findings in spQRSTa-DCM MR analyses (Supplementary Data 19). Funnel, scatter, and forest plots for HCM and DCM analyses are presented in Supplementary Figs. 8 and 9.

## Discussion

Our large-scale analyses of spQRSTa and fQRSTa -- two emerging markers for arrhythmogenesis and cardiovascular disease -- significantly advance our understanding of their basic biology and relationships with classical ECG markers. We identify candidate genes involved in cardiac development, muscle cell differentiation, cardiac contraction and actin-filament based movement. The genes also haverelationships with cardiomyopathies and central arterial vascular development. spQRSTa and fQRSTa shared loci with other ECG

measures. But there are also 7 unshared loci, suggesting distinct genetic influences. Among spQRSTa and fQRSTa loci, there are fewer genes for cardiac ion channels, in contrast to findings for other ECG traits. Based on a phenome-wide scan, we report associations with atrial fibrillation, conduction disease and arterial embolism. Moreover, PRSs are associated with fascicular and bundle branch block, and AV block, indicating potential downstream effects of the loci.

A substantial proportion of lead candidate genes at spQRSTa loci are associated with development of inherited cardiomyopathies in humans (including *MYH7, TTN, TNNT2, MYBPC3, DSP, RBM20*; Fig. 7)[34]. There was also support for genes with non-Mendelian roles in cardiac myogenesis, including *ADPRHL1, NACA* and NFIA. The function of *ADPRHL1* in humans has yet to be established, however, knockout of *ADPRHL1* in *Xenopus laevis* causes loss of the myofibril assembly in ventricular cardiomyocytes and prevents ventricular outgrowth[35].

Small clinical studies have identified an association between a widened spQRSTa and HCM in paediatric and adult populations[10,36]. A widened spQRSTa also predicts occurrence of ventricular arrhythmia among HCM patients[37]. Interestingly, we did not identify a causal relationship between genetically determined spQRSTa and HCM or DCM in MR studies. Lack of association could be due to the small sizes of the HCM and DCM cohorts. However, the analyses did identify GWS loci. Therefore, the spQRSTa may reflect functional information in

these cardiomyopathies (conditional, non-obligatory), rather than causal mechanisms for the structural phenotype. The spQRSTa may also reflect mechanisms and conditions predisposing to intermittent changes in ventricular conduction (e.g., intermittent or persistent BBB) indicating the development of cardiac memory[11,38]. This is supported by our PheWAS and PRS analyses, where we observed associations with fascicular or bundle branch block and AV block. Therefore, although we did not find a causal relationship with structural HCM or DCM phenotypes, the spQRSTa may reflect the burden of intermittent ventricular arrhythmia or conduction abnormalities occurring over time in these conditions[39].

Multiple findings support a role for angiogenesis and arterial development in modulating the spQRSTa, including candidate genes (*ALDH1A2*, *ANGPT1*, and *VAV2*), significant enrichment of GO-terms (coronary vascular development and vasculogenesis), and associations identified in PheWAS (arterial embolism, thrombosis and hypertension). *VAV2*, a candidate gene identified in Hispanic/Latino ancestry-specific analyses, is a guanine nucleotide exchange factor for Ras-related GTPases and modulates receptor-mediated angiogenic responses[40,41]. Knockout mice for this gene show signs of left ventricular hypertrophy, cardiac fibrosis and hypertension[42]. Abnormal angiogenesis influences cardiac structure and function through physiological and pathological cardiac hypertrophy, effects on tissue recovery following ischaemia, and regenerative capacity[43,44]. These processes may potentially lead to an arrhythmogenic substrate. A recent study identified an association between a widened spQRSTa and increased risk for cardioembolic and haemorrhagic stroke[45]. Our findings provide potential biological explanations for stroke associations.

Previous theoretical studies suggested that the spQRSTa reflects abnormalities of ventricular repolarization due to abnormal depolarization[3]. We identified shared genetic influences and loci overlapping with mainly PR interval and QRS duration. We also report loci that are shared across multiple ECG traits including *NFIA*, *CASQ2*, *RYR2*, *TTN*, *SCN5A*, *PITX2*, *CDKN1A*, *PLN*, *NACA* and *NDRG4* (Fig. 7, Supplementary Data 15). In comparison to results reported for QT and JT, there is less support for the involvement of cardiac potassium channels, which are important determinants of ventricular repolarization and common targets of existing anti-arrhythmics[46]. Combined with other studies, our results support an interpretation that the spQRSTa is primarily a marker of abnormal ventricular depolarization and suggests new therapies targeting depolarization should be investigated for arrhythmia prevention and management.

Despite evidence for shared effects at some loci, genetic and phenotypic correlations of spQRSTa and fQRSTa with other ECG traits are weak. Therefore, spQRSTa and fQRSTa may represent unique biology that may contribute to arrhythmic risk. There was no overlap with other ECG traits at 7 multi-ancestry spQRSTa loci. Candidate genes at these loci include: *AHNAK2*, which encodes a large nucleoprotein that localises to the Z-band region of mouse cardiomyocytes and may have a role in excitation-contraction coupling through effects on L-type voltage-gated calcium channels; *SGCG*, a component of the subsarcolemmal cytoskeleton; and *ALDH1A2*, which encodes an enzyme responsible for early embryonic retinoic acid synthesis, a process that is critical for normal cardiac and arterial development[47–50]. Another candidate gene *TAOK2*, is a protein kinase most studied for its role in dendritic spine maturation[51]. More recently, *TAOK2* has been identified in tethering the endoplasmic reticulum to microtubules. We report another locus, *MACF1*, that is also involved in microtubule organization[52,53]. Validation of these loci is required.

Although sample sizes were significantly larger for fQRSTa than for spQRSTa (134% larger), we found fewer loci and lower heritability estimates for fQRSTa. All multi-ancestry fQRSTa loci overlapped with spQRSTa loci. There were candidate genes involved in cardiac development and cardiomyopathies including *SCN5A*, *RBM20*, *PLN*, *TBX3*

and *MYO18B*. The fQRSTa represents the QRS-T angle in the frontal plane only, whereas the spQRSTa is 3-dimensional. Therefore the fQRSTa trait likely loses information that resides in other planes. However, we identified an unreported locus in African ancestry-specific analyses (candidate gene *FAM135B*). Knockdown of *FAM135B* in iPSC lines reduces spinal motor neuron survival and contributes to neurite defects as seen in spinal and bulbar muscular atrophy. These disorders are associated with cardiac arrhythmia and structural abnormalties[54–56].

Although our study includes individuals from multiple ancestries, ancestry-specific analyses were limited by sample sizes. Larger studies are needed to yield additional signals. The precise algorithms used to calculate the spQRSTa will marginally differ despite efforts to harmonise approaches; however, such differences are unlikely to affect our positive findings (measurement error or noise will dampen signals), and summary statistics for spQRSTa across all studies are broadly similar (Supplementary Data 3)[57–59].

In summary, our analyses significantly advance our knowledge of the underlying biology reflected by the spQRSTa and fQRSTa, which are independent risk markers for arrhythmogenesis. We also identified loci that have not been reported for ECG traits. Our findings highlight biological processes and candidate genes that may explain associations observed in previous clinical studies and could inform future research on the utility of these markers in risk prediction.

## Methods
### Study cohorts
Fourteen studies (32 ancestry-specific sub-studies) and 23 studies (40 ancestry-sub-studies) contributed GWAS summary statistics for spQRSTa and fQRSTa meta-analyses, respectively. These included members of the Cohorts for Heart and Aging Research in Genomic Epidemiology (CHARGE) consortium[60] (Supplementary Data 1). This study was approved by all participating cohorts. Ethics and consent was obtained at a study level. The majority of participating cohorts were population based with a small number of case-control studies. Information for study level genotyping method (typically Illumina or Affymetrix), quality control (Hardy-Weinberg equilibrium [HWE] *P*, and MAF), are provided in Supplementary Data 2. The 1000 G reference panel (26/40 sub-studies) was most used for imputation (26/40) followed by the Haplotype Reference Consortium panel (13/40). The Atherosclerosis Risk in Communities (ARIC) study was imputed with TOPMed Freeze 5 reference panel[61,62]. All GWAS summary data included in the meta-analyses utilized NCBI build 37 (summary statistics for ARIC sub-studies were converted from build 38 to 37 using a liftover tool [https://genome.sph.umich.edu/wiki/LiftOver]).

### Cohort-level single variant association analyses
A GWAS was performed by each participating cohort for the spQRSTa (mean) and fQRSTa. If the spQRSTa was not already calculated and digitized ECGs were available, it was derived by transformation of the 12-lead ECG using previously published algorithms[57]. In brief, after applying a bandpass butterworth filter and signal averaging to reduce noise, orthogonal X, Y and Z vector beats were estimated using Kors' regression matrix[63]. The spQRSTa was subsequently calculated as the angle between mean QRS and T-wave spatial vector loops[57,58]. The fQRSTa was defined as the absolute difference between QRS and T-wave frontal plane axes ($fQRSTa = abs[QRS\text{-}axis - T\text{-}axis]$)[3]. Values for both phenotypes are between 0 and 180°.

The primary analysis for this study to declare GWS and previously unreported associations, was the rank-based inverse normal transformed phenotype (as both the spQRSTa and fQRSTa are not normally distributed). The raw phenotype was also analysed to calculate clinically meaningful effect sizes (on the degree [°] scale). Study level GWASs were performed using an additive genetic model, accounting for relatedness with appropriate software (e.g. BOLT linear mixed

model [BOLT-LMM])[15] or by including a kinship matrix or pedigree[64–66]. Poorly imputed genotypes were excluded (Rsq > 0.3 or similar for IMPUTE) and a MAF > 0.01 was applied, so that only high-quality variants were included in the study.

Summary statistics for cohort level distributions of each ECG trait and covariates included in the GWAS model, are provided in Supplementary Data 3. Age, sex, RR interval, height, and body-mass index (BMI) were mandatory covariates in the GWAS model. In addition, as the QT interval is associated with the QRS-T angle and we wished to identify associations that were not primarily driven by this marker of ventricular depolarization and repolarization, the QT interval was also included as a covariate. If pedigree data was not available, or if the chosen GWAS software did not correct for underlying population stratification, genetic principal components (PCs) were also included as covariates. Cohorts could also select additional covariates if relevant to their study, such as genotyping method or recruitment site. Cohorts comprising multiple ancestries performed separate analyses for each ancestry.

Individuals were not included in the study-level GWAS if they had a prior diagnosis of heart failure, myocardial infarction, pacemaker or implantable cardiac defibrillator; were prescribed class I or III anti-arrhythmics, QT-prolonging or digitalis medication; or were pregnant at the time of ECG acquisition. In addition, individuals were excluded if atrial fibrillation, BBB or a QRS duration greater than 120 ms, was present on their ECG.

## Additional quality control of study-level data

After submissions of results in a standardized format, quality control was performed using EasyQC (R package v9.2)[67]. Allele frequencies of all variants were compared to those reported in the reference panel used by the study for imputation. To identify analytical errors, QQ and P–Z-score plots were inspected, and summary statistics for β estimates and SE were compared across all studies. To identify potential uncorrected population stratification, the genomic-control inflation factor was calculated to identify test statistic inflation.

## GWAS meta-analysis

The primary GWAS meta-analysis for spQRSTa and fQRSTa was the multi-ancestry rank-based inverse normal transformed meta-analysis; however, to estimate clinically relevant effect sizes, a GWAS meta-analysis was also performed using the untransformed phenotype (on the degree [°] scale). European, African, and Hispanic/Latino ancestry-specific meta-analyses were also performed as secondary analyses. Meta-analyses were performed with METAL (v2011-03-25) using an IVW model[68]. If a study's λ was >1.0, genomic control during the meta-analysis. Summary statistics and plots were produced for the entire meta-analysis. Subsequently in downstream analyses, variants were only included if present in >60% of the total meta-analysis sample size. The GWS threshold was set as $P < 5 \times 10^{-08}$. To calculate the correlation between variants, relevant individuals from the 1000 G reference panel were used; all individuals for the multi-ancestry summary statistics, ancestry-specific for European, African and Hispanic/Latino analyses. Some in-silico analyses relied upon correlations calculated by the software developers and did not permit modification. In these instances (and explicitly stated in the manuscript text), only European-ancestry summary statistics were used in recognition that the multi-ancestry meta-analysis contained a substantial proportion of individuals of European descent.

## Definition of known and previously unreported loci

One previous GWAS has been reported for spQRSTa, with 3 loci reaching GWS[13]. Using PLINK (v1.9)[69], lead variants from the study were extracted to calculate locus boundaries, defined as ±500 kb or r2 < 0.1 within a 4 mb region (whichever was larger), centered on the lead variant. The 1000 G reference panel was used to calculate correlations

between variants[61]. The variants furthest upstream or downstream were declared the locus start and end respectively. We used the same list to define known loci for fQRSTa as no previous GWAS has been reported for this trait and as the phenotypic correlation with spQRSTa is high. The same method was used to identify GWS loci in our study. Loci that did not overlap with the list of known loci, were declared as previously unreported.

Heterogeneity I[2] statistics and forest plots were produced for each lead variant (smallest P) at each locus, to identify evidence for heterogeneity. Locus-Zoom plots were also produced to visually inspect patterns of association and variant correlations[70].

## Conditional and heritability analyses

To identify independent secondary signals within GWS loci, conditional analyses using European-ancestry statistics were performed using Genome-wide Complex Trait Analysis (GCTA, v1.26.0)[14]. As recommended by GCTA, the largest cohort in the meta-analysis was used as the reference panel (UKB, N = 33,960). For this analysis, related individuals in the UKB sample were excluded (up to the 2nd-degree [kinship coefficient <0.0884]). A strict threshold (r² < 0.1 with the lead variant and $P_{Joint} < 5 \times 10^{-08}$) was used to declare "conditionally independent" signals within loci.

Heritability estimates were calculated using the same UKB individuals of European-ancestry included in the GWAS meta-analysis, using BOLT-REML (v2.3.2)[15]. BOLT-REML models directly genotyped SNPs to estimate relatedness within a sample and obtains SNP-based heritability estimates. The percentage of variance explained (PVE) by lead and conditionally independent variants was subsequently calculated (Eq. 1)[71];

$$PVE = \frac{[2*(beta\hat{\ }2)*MAF*(1-MAF)]}{[2*(beta\hat{\ }2)*MAF(1-MAF) + ((se(beta))\hat{\ }2)*2*N*MAF*(1-MAF)]} \quad (1)$$

## Variant annotation

Lead and conditionally independent variants (and their proxies [r² > 0.8]) were annotated using Variant Effect Predictor (VEP, Ensembl release 99) to identify potential functional consequences[16]. VEP also contains data from prediction tools Sorting Intolerant From Tolerant algorithm (SIFT, version 5.2.2)[72] and PolyPhen-2 (v2.2.2[73]), which supplied deleteriousness scores. In addition, CADD[74] and RegulomeDB[75] scores for each of these variants were extracted. CADD scores annotate coding and non-coding variants, and enable ranking of their potential deleteriousness compared with other variants in the genome[74].

## Association with tissue-specific gene expression

To identify relationships between lead and conditionally independent variants (and their proxies), with tissue-specific gene expression, cis-eQTL data was extracted from the GTEx portal (v8)[17,76,77]. Tissues included in these analyses were cardiac (LV and RAA) and vascular (coronary and aorta artery), for their known influence on cardiovascular disease. If a variant was also a lead cis-eQTL variant, colocalization analysis were performed at the locus using the R package COLOC, to determine whether the variant was causal in both the GWAS meta-analysis and the eQTL study[78]. These colocalization analyses use Bayesian statistical methods to calculate a posterior probability (PP) for the variant being causal in both analyses (PP > 75%).

To predict the effects of gene expression levels on spQRSTa and fQRSTa, we performed a TWAS using S-PrediXcan. S-PrediXcan is an extension of the original software PrediXcan and infers results using GWAS summary statistics, thus removing the need for individual-level genotype and phenotype data[79]. S-PrediXcan provides a precalculated transcriptome model database from GTEX-based tissues and covariance matrices of SNPs within each gene model (https://github.com/hakyimlab/MetaXcan). We used European meta-analysis summary

statistics for these analyses and tested for association in a total of 16,097 genes across LV, RAA and vascular tissues. A Bonferroni corrected threshold (0.05/number of genes tested [16,097] = $3.1 \times 10^{-6}$) was used to declare significance and results are only reported when more than one SNP was included in the model.

## Tissue- and cell-type specific regulatory elements

GARFIELD (v2) was used to identify tissue-specific enrichment of variants at DNase I hypersensitivity sites[80]. GARFIELD annotates variants with data from the ENCODE, GENCODE and Roadmap Epigenomics projects and calculated odd ratio using a generalised linear model framework[80].

Chromatin interaction data was used to identify target genes of regulatory variants (RegulomeDB score ≤3b) in LV and RV tissues. First, using FUMA GWAS (Functional Mapping and Annotation of Genome-Wide Association Studies) software (v1.3.6), overlap was identified between lead and conditionally independent variant, and pre-processed loops determined by Fit-Hi-C pipelines[18,81]. An FDR threshold <0.05 was used to report results. In addition, we performed the same analysis using loops called from recently published Knight-Ruiz normalised 5 kb, 10 kb and 25 kb resolution promotor capture Hi-C data[19].

To identify cardiac cell-type specific enrichment of non-coding variants, we utilized accessible chromatin information from snATAC-seq data, for atrial and ventricular cardiomyocyte, smooth muscle, endothelial, adipocyte, macrophage, fibroblast, lymphocyte and nervous cells[21]. Using PLINK, our GWAS meta-analysis summary statistics were partitioned into haplotype blocks centered on each lead variant ($r^2 > 0.1$ within a 2 Mb radius). Peaks within the lowest decile of total read counts from the snATAC-seq data were removed using a SNP enrichment method CHEERS (version accessed 2020)[20], followed by quantile normalization of the remaining peak counts[20]. Enrichment of variants (one-sided $P$) within the ATAC-seq peaks was estimated and a Bonferroni-corrected threshold (0.05/number of cell-types) used to report significant findings.

## Candidate gene prioritisation and pathway enrichment

To identify additional candidate genes at each locus, DEPICT (v3) software was used, that prioritizes genes according to common functional pathways. DEPCT calculates a membership probability for each gene within 14,461 reconstituted gene-sets[22]. Additional analyses were performed using DEPICT to identify pathway enrichment of these genes using Gene-Ontology (GO), Kyoto Encyclopaedia of Genes and Genomes (KEGG), REACTOME and the Mouse genetics initiative (MGI) data. DEPICT also performs gene-set tissue enrichment analyses using annotations from human Affymetrix microarray probes. For all analyses, an FDR < 0.01 was used to identify significant results. To visualise GO biological processes from the DEPICT spQRSTa multi-ancestry meta-analysis output were analysed using the REVIGO web application to remove redundant terms and cluster related nodes[23]. They were subsequently visualised using Cytoscape (v3.8.2)[82].

The output of all bioinformatic analyses were pooled and supplemented with trait relevant information from Online Mendelian Inheritance in Man (OMIM)[24] and International Mouse Phenotyping Consortium[25] (IMP, www.mousephenotype.org) databases, the Human Protein Atlas[26] (www.proteinatlas.org) and a PubMed literature review (Supplementary Data 14). We also performed a look up of each lead variant in the Open Targets Genetics "Locus to Gene" machine learning pipeline, which uses supervised learning to weight evidence from different sources and prioritize genes at a locus[27]. This database does not include trait-specific information in the pipeline and therefore it is used to supplement the analyses performed for this work. For each locus, the candidate gene with the most support across all lines of evidence is indicated. We also included a second gene if there is support from multiple analyses.

## LD score regression

LD score regression with LDSC (v1.0.1), was performed to calculate the genetic correlation of the spQRSTa and fQRSTa with other ECG traits including PR, QRS, JT and QT intervals[28]. LDSC (v1.0.1) uses pre-computed LD scores and therefore these analyses were performed with European ancestry summary statistics only. These LD scores are used as weights in the regression model[29].

## Overlap of spQRSTa loci with other resting ECG traits and association with clinical phenotypes

Lead variants previously reported for other ECG markers including P-wave duration, atrioventricular conduction (PR interval[12]), ventricular depolarization (QRS duration[83] and QRS voltage[84]), ventricular repolarization (JT[83], QT[83] and Tp-Tend intervals[85]) and HR[86] were tested for overlap with spQRSTa loci (definition of overlap; if previously reported lead variants were within ±500 kb or $r^2 > 0.1$ of the lead spQRSTa variant). Summary statistics for each ECG trait were also extracted and pairwise-GWASs performed using Bayesian bivariate analyses as implemented in GWAS-PW[30]. GWAS-PW combines GWAS summary statistics using the variance of effect sizes at each SNP to estimate the probability that a given genomic region contains a variant that influences both traits or distinct associations, and learns reasonable thresholds from the data to declare significance. A pairwise GWAS was performed with the summary statistics of the multi-ancestry spQRSTa meta-analysis and each ECG trait of interest. To account for sample overlap between summary statistics, an expected correlation (-cor in GWAS-PW) between two traits was specified for each analysis[87]. The values used after adjusted for estimated sample overlap were; −0.0045, −0.0258, 0.0539, 0.0107, 0.0045 and 0.0135 for QT, JT, QRS, PR, HR and TpTe respectively. A posterior probability >0.9 was used as evidence supporting the presence of a causal SNP within the genomic region that influences both traits.

To identify evidence of pleiotropy with clinical conditions, a PheWAS was performed using the R package PheWAS (v0.99.5-5)[88]. ICD-10 and 9 codes were extracted from UKB hospital episode statistics and mapped to phecodes. Lead and conditionally independent variants from the European ancestry spQRSTa meta-analysis were subsequently tested for association with each phecode in 395,758 European individuals. Related pairs were excluded (kinship coefficient >0.0884). A Bonferroni corrected threshold for the number of phecodes tested (0.05/1,301 = $3.8 \times 10^{-5}$) was used to declare significance. To identify evidence for pleiotropy with non-cardiac phenotypes and diseases from previously reported GWAS, a look-up was performed of lead and conditionally-independent spQRSTa variants (and proxies, $r^2 > 0.8$) using Phenoscanner v2[89,90]. Associations reaching GWS with other traits and diseases were extracted.

## Sensitivity analyses

To determine whether the QT interval significantly influences the findings from our spQRSTa meta-analyses, sensitivity analyses were performed in UKB ($N = 34,361$). Analyses were repeated without the QT interval as a covariate. Spearman rank correlations (rho [$\rho$]) for beta estimates and -log10 $P$-values, were calculated across all variants between the original model and the sensitivity analysis.

## Association between genetically determined spQRSTa and fQRSTa with cardiovascular diseases

A PRS was calculated for each trait using lead variants from the European meta-analysis, to test for association with atrial fibrillation, stroke, coronary artery disease, conduction disease, heart failure, non-ischaemic cardiomyopathy and ventricular arrhythmia. Analyses were performed in individuals of European ancestry in UKB ($N = 395,758$). Participants included in the GWAS meta-analysis were excluded, along with related pairs up to the 2nd-degree (kinship coefficient <0.0884). To take advantage of genotype probability data in BGEN format,

PRSice-2[91] was used. The PRSs were calculated by summing the dosage of lead variants weighted by the effect size from the corresponding raw-phenotype meta-analysis. Disease status for each cardiovascular outcome of interest was extracted using ICD-9/ICD-10 codes from hospital admission episodes, self-reported data, operation codes and death certification (Supplementary Note 2). Associations were identified for prevalent and incident cases using a logistic regression model, including covariates age, sex, genotyping array and ten genetic principal components. A Bonferroni-corrected threshold of 0.05/number of outcomes tested $(0.05/8 = 6.3.1 \times 10^{-3})$ was used to declare significant associations.

### Relationship between spQRSTa and HCM and DCM

The TwoSampleMR R package (v0.5.6), was performed to test for association of spQRSTa with cardiomyopathies, using data from cohorts with HCM and DCM[92]. First, summary statistics from a previously reported multi-ancestry (2780 cases, 47,486 controls) and European (2,244 cases, 42,668 controls) HCM GWAS were retrieved[32]. Summary statistics for multi-ancestry sarcomere positive (871 cases, 20,142 controls) and sarcomere negative (1874 cases, 27,344 controls) HCM GWAS were also extracted. The HCM GWAS included UKB participants as controls; however, there was no overlap of individuals included in the spQRSTa meta-analyses. Multi-ancestry (61 variants) and European (51 variants) IVs were constructed from GWS variants in the rank-based inverse normal transformed spQRSTa meta-analysis, with the corresponding β, SE and $P$ retrieved from the untransformed meta-analysis to facilitate clinical interpretation. Effect alleles were harmonised between IVs and HCM summary statistics. Two variants, rs398110577 and rs35185344, from the multi-ancestry IV were unavailable within the HCM summary statistics and proxies were selected, rs4946230 ($r^2 = 0.70$) and rs12928779 ($r^2 = 0.98$), respectively. Four different methods were performed, specifically IVW, MR-Egger, weighted median and MR-PRESSO (mendelian randomisation pleiotropy residual sum and outlier), using MR-Base[92,93]. Results are reported as OR (95% CI) for risk of HCM per 1° increase in genetically determined spQRSTa. The same process was followed to test for association with DCM, but with the following differences. Summary statistics from a European ancestry "sporadic" DCM GWAS (2651 cases, 4329 controls) were used[33]. Sporadic DCM was defined as a reduced LV ejection fraction and enlarged LV end-diastolic volume/diameter in the absence of any obvious pathology[33]. For these analyses, one variant from the European IV was not available (rs2668692), therefore a suitable proxy was selected (rs10514897, $r^2 = 0.78$).

### Reporting summary

Further information on research design is available in the Nature Portfolio Reporting Summary linked to this article.

## Data availability

Summary statistics from each genome-wide association study meta-analysis have been uploaded to the NHGRI-EBI Catalog of human genome-wide association studies website, https://www.ebi.ac.uk/gwas/.(Study accession numbers GCST90246318, GCST90246320, GCST90246322, GCST90246324 for Multi-ancestry, European, African and Hispanic ancestry meta-analyses for the spatial QRS-T angle, respectively. Study accession numbers GCST90246319, GCST90246321, GCST90246323, GCST90246325 for Multi-ancestry, European, African and Hispanic ancestry meta-analyses for the frontal QRS-T angle, respectively). Data relating to UK Biobank will be return to the study. The UK Biobank will make these data available to all bona fide researchers for all types of health-related research that is in the public interest, without preferential or exclusive access for any person. All researchers will be subject to the same application process and approval criteria as specified by the UK Biobank. Please see the UK Biobank's website for the detailed access procedure (http://www.ukbiobank.ac.uk/register-apply/). Other datasets used in these analyses are publicly available and can be sourced from: 1000 Genomes reference panel: https://www.internationalgenome.org/category/reference/; Haplotype reference consortium reference panel: http://www.haplotype-reference-consortium.org/; Variant level annotation from Variant Effect Predictor (VEP), Ensembl release 99: https://www.ensembl.org/info/docs/tools/vep/index.html; Variant level Combined Annotation Dependent Depletion scores from Combined Annotation Dependent Depletion (CADD, v1.4): https://cadd.gs.washington.edu/; Variant level tissue-specific gene expression from The GTEx portal (v8): https://gtexportal.org/home/; HiC data from the Functional Mapping and Annotation of Genome-Wide Association Studies (FUMA GWAS, v.1.3.6): https://fuma.ctglab.nl/; DNaseI hypersensity site enrichment data from GWAS Analysis of Regulatory and Functional Information Enrichment with LD correction (GARFIELD, v2): https://www.ebi.ac.uk/birney-srv/GARFIELD/; Gene-set, biological pathways and tissue expression data from Data-driven Expression-Prioritization Integration for Complex Traits (DEPICT, v3): https://github.com/perslab/depict; Variant level RegulomeDB scores from RegulomeDB (v.2.0.3): https://regulomedb.org/regulome-search/; A compendium of promoter-centered long-range chromatin interactions in the human genome (Jung et al., 2019): https://doi.org/10.1038/s41588-019-0494-8; Cardiac cell type-specific gene regulatory programs and disease risk association (Hocker et al., 2021): DOI: 10.1126/sciadv.abf1444; Druggable genome dataset from Finan et al., 2017: DOI: 10.1126/scitranslmed.aag1166; g:Profiler (accessed May 2021): https://biit.cs.ut.ee/gprofiler/gost; Online Mendelian Inheritance in Man database: https://www.omim.org/; Mouse Genome Informatics: http://www.informatics.jax.org/; KEGG drug database: (https://www.genome.jp/).

## Code availability

Codes are available from the original software used for each analysis.

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

## Acknowledgements

All study acknowledgements can be found in Supplementary Note 3, and study funding information in Supplementary Note 4. W.J.Y acknowledges support by an MRC grant MR/R017468/1. A.A is supported by an NHLBI award K24HL148521. A.L.P.R is supported in part by CNPq (310679/2016-8 and 465518/2014-1) and by FAPEMIG (PPM-00428-17 and RED-00081-16). E.T-S is supported by Brazilian Ministry of Health (National Program of Genomics and Precision Health), Brazilian Conselho Nacional de Desenvolvimento Científico e Tecnologico (CNPq), Fundação de Amparo a Pesquisa do Estado de Minas Gerais (FAPEMIG, RED00314-16). M.F.L-C is supported by Brazilian Ministry of Health (DECIT/MS, EPIGEN-Brazil Project), Brazilian Ministry of Science and Technology (Financiadora de Estudos e Projetos (FINEP), Brazilian Conselho Nacional de Desenvolvimento Científico e Tecnologico (CNPq), Fundação de Amparo a Pesquisa do Estado de Minas Gerais (FAPEMIG)). M.L.S is supported by Brazilian Conselho Nacional de Desenvolvimento Científico e Tecnologico (CNPq), Third- World

Academy of Science (TWAS), Fogarty International Center of the US National Institutes of Health (D43 TW007393). P.B.M, A.T, P.D.L and M.O acknowledge support by an MRC grant MR/N025083/1. P.B.M, H.R.W, A.T and P.D.L acknowledge the NIHR Barts Biomedical Research Centre at Queen Mary University of London. P.D.L is also supported by UCL/UCLH Biomedicine NIHR, Barts Heart Centre Biomedical Research Centre. J.R acknowledges support from the European Union's Horizon 2020 research and innovation programme under the Marie Sklodowska-Curie grant agreement No.786833, and the "María Zambrano" fellowship through the European Union-NextGenerationEU. N.S is supported by grants AHA19SFRN348300063, R01HL141989, Medic One Foundation, Laughlin Family. U.S received funding from the Netherlands Heart Foundation (CVON2014-09, RACE V Reappraisal of Atrial Fibrillation: Interaction between hyperCoagulability, Electrical remodeling, and Vascular Destabilization in the Progression of AF), the European Union (ITN Network Personalize AF: Personalized Therapies for Atrial Fibrillation: a translational network, grant number 860974; MAESTRIA: Machine Learning Artificial Intelligence Early Detection Stroke Atrial Fibrillation, grant number 965286). S.T is supported by a Junior 1 Clinical Research Scholar award from the Fonds de Recherche du Québec-Santé (FRQS). J.L.I is supported by CACHET. M.O was supported by The John and Birthe Meyer Foundation and The Hallas-Møller Emerging Investigator Novo Nordisk (NNF17OC0031204). C.H is supported by an MRC University Unit Programme Grant MC_UU_00007/10 (QTL in Health and Disease). C.A and A.B were supported by NIH grants R01HL142825, and U01HG007416. D.D was supported by NIH grants R01HL138737 and T32HL139439. N.G, C.N and T.H are supported by the Novo Nordisk Foundation (Grant number NNF18CC0034900). H.M is supported by the CHARGE infrastructure grant (HL105756). J-W.B is funded by the Research Project CVON-AI (2018B017) financed by the PPP Allowance made available by Top Sector Life Sciences & Health to the Dutch Heart Foundation to stimulate public-private partnerships. D.M-K is supported by Dutch Science Organization (ZonMW-VENI Grant 916.14.023). J.F.W acknowledges support from the MRC Human Genetics Unit programme grant, "Quantitative traits in health and disease" (U. MC_UU_00007/10). L.R is funded by a University of Edinburgh studentship. P.N is supported by the MRC Human Genetics Unit programme grant, "Quantitative traits in health and disease" (U. MC_UU_00007/10). X.S was in receipt of a Starting Grant (2017-02543) from the Swedish Research Council (Vetenskaprådet). This research has been conducted using the UK Biobank Resource under Application Number 8256. This research used data assets made available by National Safe Haven as part of the Data and Connectivity National Core Study, led by Health Data Research UK in partnership with the Office for National Statistics and funded by UK Research and Innovation (grant ref MC_PC_20029). Copyright © (2022), NHS Digital. Re-used with the permission of the NHS Digital [and/or UK Biobank]. All rights reserved. The authors also wish to acknowledge the CHARGE infrastructure grant (HL105756).

## Author contributions

Interpreted results, writing, and editing the manuscript: W.J.Y, L.T, P.B.M. Conceptualization of project: W.J.Y, N.S, L.T, P.B.M. Supervision of project: L.T, P.B.M. Contributed to GWAS analysis plan: W.J.Y, N.S, L.T, P.B.M. Performed meta-analyses: W.J.Y and J.Ra. Performed GCTA, genetic correlations, heritability, variant annotations, GTEx analyses, HiC analyses, gene-set enrichment and pathway analyses, PheWAS, TWAS, Pairwise GWAS, Gene literature review, Phenoscanner look up, Comparison with existing ECG loci, PRS analyses in UKB: W.J.Y. Performed MR analyses using HCM GWAS meta-analysis summary statistics and provided text for the relevant section: A.R.H, A.G, C.G, H.Wat. Contributed data for DCM-spQRSTa MR analysis: S.G, J.-F.D, E.V, P.C. Provided support for bioinformatic analyses: B.M. Performed heritability analyses in ERF: A.I. Contributed to study-specific GWAS by providing phenotype, genotype and performing data analyses: W.J.Y, J.H, J.-W.B, L.Re, J.Y, A.I, J.Ra, S.V.D, A.R.B, M.P.C, T.D, L.F, J.L.I, H.M, R.N, C.N, A.Ric, M.Sa, C.M.S, N.Sor, S.Th, S.Tr, S.A, F.A, A.A, J.A.B, A.Ca, A.Co, D.D, A.D.L, C.E, C.F, X.G, T.H, S.R.H, R.D.J, J.A.K, M.F.L-C, A.L, P.W.M, A.C.M, P.N, D.J.P, P.P.P, A.Rei, L.Ri, U.S, X.S, G.S, E.Z.S, M.St, E.T-S, A.T, K.T, H.War, E.A.W, K.L.W, D.E.A, C.L.A, D.C, G.G, N.G, C.H, J.W.J, D.O.M-K, M.Ol, S.P, B.M.P, C.P, A.Rib, J.Ro, B.S, P.H, C.M.D, N.V, J.G.W, M.Or, C.K, H.J.L, J.F.W, J.K.K, N.Sot, P.D.L, P.B.M. All authors read, revised, and approved the manuscript.

## Competing interests

B.M.P serves on the Steering Committee of the Yale Open Data Access Project funded by Johnson & Johnson. D.C has received speaker fees from BMS/Pfizer and Servier, and consultation fees from Roche Diagnostics and Trimedics. U.S received consultancy fees or honoraria from Università della Svizzera Italiana (USI, Switzerland), Roche Diagnostics (Switzerland), EP Solutions Inc. (Switzerland), Johnson & Johnson Medical Limited, (United Kingdom), Bayer Healthcare (Germany). D.O.M.-K is a part time research consultant at Metabolon, Inc. U.S is co-founder and shareholder of YourRhythmics BV, a spin-off company of the University Maastricht. The remaining authors declare no competing interests.

## Additional information

William J. Young [1,2], Jeffrey Haessler[3], Jan-Walter Benjamins[4], Linda Repetto[5], Jie Yao[6], Aaron Isaacs[7,8], Andrew R. Harper[9,10], Julia Ramirez[1,11,12], Sophie Garnier[13,14], Stefan van Duijvenboden[1,11], Antoine R. Baldassari[15], Maria Pina Concas[16], ThuyVy Duong[17], Luisa Foco[18], Jonas L. Isaksen[19], Hao Mei[20], Raymond Noordam[21], Casia Nursyifa[22], Anne Richmond[23], Meddly L. Santolalla[24,25], Colleen M. Sitlani[26], Negin Soroush[27], Sébastien Thériault[28,29], Stella Trompet[21,30], Stefanie Aeschbacher[31], Fariba Ahmadizar[27,32], Alvaro Alonso[33], Jennifer A. Brody[26], Archie Campbell[34,35,36], Adolfo Correa[37], Dawood Darbar[38], Antonio De Luca[39], Jean-François Deleuze[40,41,42], Christina Ellervik[43,44,45], Christian Fuchsberger[18,46,47], Anuj Goel[9,10], Christopher Grace[9,10], Xiuqing Guo[6,48,49], Torben Hansen[22], Susan R. Heckbert[26,50], Rebecca D. Jackson[51], Jan A. Kors[52], Maria Fernanda Lima-Costa[53], Allan Linneberg[54,55], Peter W. Macfarlane[56], Alanna C. Morrison[57], Pau Navarro[23], David J. Porteous[36,58], Peter P. Pramstaller[18,59], Alexander P. Reiner[50,60], Lorenz Risch[61,62,63], Ulrich Schotten[7], Xia Shen[5,64,65], Gianfranco Sinagra[39], Elsayed Z. Soliman[66], Monika Stoll[8,67,68], Eduardo Tarazona-Santos[24], Andrew Tinker[1,69], Katerina Trajanoska[70], Eric Villard[13,14], Helen R. Warren[1,69], Eric A. Whitsel[15,71], Kerri L. Wiggins[26], Dan E. Arking[17], Christy L. Avery[15], David Conen[28], Giorgia Girotto[16,72], Niels Grarup[22], Caroline Hayward[73], J.Wouter Jukema[30,74,75], Dennis O. Mook-Kanamori[76,77], Morten Salling Olesen[78], Sandosh Padmanabhan[79], Bruce M. Psaty[26,50,80], Cristian Pattaro[18], Antonio Luiz P. Ribeiro[81,82], Jerome I. Rotter[6,48,83], Bruno H. Stricker[27], Pim van der Harst[4,84], Cornelia M. van Duijn[85,86], Niek Verweij[4], James G. Wilson[87,88], Michele Orini[2,11], Philippe Charron[13,14,89,90], Hugh Watkins[9,10], Charles Kooperberg[3], Henry J. Lin[6,48,49], James F. Wilson[5,23], Jørgen K. Kanters[19], Nona Sotoodehnia[91], Borbala Mifsud[1,92], Pier D. Lambiase[2,11], Larisa G. Tereshchenko[93,94,95] ✉ & Patricia B. Munroe[1,69,95] ✉

[1]William Harvey Research Institute, Clinical Pharmacology, Queen Mary University of London, London, UK. [2]Barts Heart Centre, St Bartholomew's Hospital, Barts Health NHS trust, London, UK. [3]Public Health Sciences Division, Fred Hutchinson Cancer Center, Seattle, WA, USA. [4]University of Groningen, University Medical Center Groningen, Department of Cardiology, Groningen, the Netherlands. [5]Centre for Global Health Research, Usher Institute, University of Edinburgh, Edinburgh, Scotland. [6]Institute for Translational Genomics and Population Sciences/The Lundquist Institute at Harbor-UCLA Medical Center, Torrance, CA, USA. [7]Dept. of Physiology, Cardiovascular Research Institute Maastricht (CARIM), Maastricht University, Maastricht, the Netherlands. [8]Maastricht Center for Systems Biology (MaCSBio), Maastricht University, Maastricht, the Netherlands. [9]Radcliffe Department of Medicine, University of Oxford, Division of Cardiovascular Medicine, John Radcliffe Hospital, Oxford, UK. [10]Wellcome Centre for Human Genetics, Roosevelt Drive, Oxford, UK. [11]Institute of Cardiovascular Sciences, University of College London, London, UK. [12]Aragon Institute of Engineering Research, University of Zaragoza, Zaragoza, Spain and Center of Biomedical Research Network, Bioengineering, Biomaterials and Nanomedicine, Zaragoza, Spain. [13]Sorbonne Universite, INSERM, UMR-S1166, Research Unit on Cardiovascular Disorders, Metabolism and Nutrition, Team Genomics & Pathophysiology of Cardiovascular Disease, Paris 75013, France. [14]ICAN Institute for Cardiometabolism and Nutrition, Paris 75013, France. [15]Department of Epidemiology, Gillings School of Global Public Health, University of North Carolina at Chapel Hill, Chapel Hill, NC, USA. [16]Institute for Maternal and Child Health – IRCCS "Burlo Garofolo", Trieste, Italy. [17]McKusick-Nathans Institute, Department of Genetic Medicine, Johns Hopkins University School of Medicine, Baltimore, MD, USA. [18]Eurac Research, Institute for Biomedicine (affiliated with the University of Lübeck), Bolzano, Italy. [19]Laboratory of Experimental Cardiology, Department of Biomedical Sciences, University of Copenhagen, Copenhagen, Denmark. [20]Department of Data Science, University of Mississippi Medical Center, Jackson, MS, USA. [21]Department of Internal Medicine, section of Gerontology and Geriatrics, Leiden University Medical Center, Leiden, the Netherlands. [22]Novo Nordisk Foundation Center for Basic Metabolic Research, Faculty of Health and Medical Sciences, University of Copenhagen, Copenhagen, Denmark. [23]MRC Human Genetics Unit, Institute of Genetics and Cancer, University of Edinburgh, Edinburgh, Scotland. [24]Department of Genetics, Ecology and Evolution, Instituto de Ciências Biológicas, Universidade Federal de Minas Gerais, Belo Horizonte, Minas Gerais, Brazil. [25]Emerge, Emerging Diseases and Climate Change Research Unit, School of Public Health and Administration, Universidad Peruana Cayetano Heredia, Lima 15152, Peru. [26]Cardiovascular Health Research Unit, Department of Medicine, University of Washington, Seattle, WA, USA. [27]Department of Epidemiology, Erasmus Medical Center, Rotterdam, the Netherlands. [28]Population Health Research Institute, McMaster University, Hamilton, ON, Canada. [29]Department of Molecular Biology, Medical Biochemistry and Pathology, Université Laval, Quebec, QC, Canada. [30]Department of Cardiology, Leiden University Medical Center, Leiden, the Netherlands. [31]Cardiovascular Research Institute Basel, University Hospital Basel, University of Basel, Basel, Switzerland. [32]Julius Global Health, University Utrecht Medical Center, Utrecht, the Netherlands. [33]Department of Epidemiology, Rollins School of Public Health, Emory University, Atlanta, GA, USA. [34]Usher Institute, University of Edinburgh, Nine, Edinburgh Bioquarter, 9 Little France Road, Edinburgh, UK. [35]Health Data Research UK, University of Edinburgh, Nine, Edinburgh Bioquarter, 9 Little France Road, Edinburgh, UK. [36]Centre for Genomic and Experimental Medicine, Institute of Genetics and Cancer, University of Edinburgh, Western General Hospital, Edinburgh, UK. [37]Departments of Medicine, Pediatrics and Population Health Science, University of Mississippi Medical Center, Jackson, MS, USA. [38]Department of Medicine, University of Illinois at Chicago, Chicago, IL, USA. [39]Cardiothoracovascular Department, Division of Cardiology, Azienda Sanitaria Universitaria Giuliano Isontina and University of Trieste, Trieste, Italy. [40]Université Paris-Saclay, CEA, Centre National de Recherche en Génomique Humaine (CNRGH), 91057 Evry, France. [41]Laboratory of Excellence GENMED (Medical Genomics), Paris, France. [42]Centre d'Etude du Polymorphisme Humain, Fondation Jean Dausset, Paris, France. [43]Department of Data and Data Support, Region Zealand, 4180 Sorø, Denmark. [44]Department of Clinical Medicine, Faculty of Health and Medical Sciences, University of Copenhagen, 2100 Copenhagen, Denmark. [45]Department of Laboratory Medicine, Boston Children's Hospital, Harvard Medical School, 300 Longwood Avenue, Boston, MA 02115, USA. [46]Department of Biostatistics, University of Michigan School of Public Health, Ann Arbor, MI, USA. [47]Center for Statistical Genetics, University of Michigan School of Public Health, Ann Arbor, MI, USA. [48]Department of Pediatrics, Harbor-UCLA Medical Center, Torrance, CA, USA. [49]Department of Pediatrics, David Geffen School of Medicine at UCLA, Los Angeles, CA, USA. [50]Department of Epidemiology, University of Washington, Seattle, WA, USA. [51]Center for Clinical and Translational Science, Ohio State Medical Center, Columbus, OH, USA. [52]Department of Medical Informatics, Erasmus University Medical Center, Rotterdam, the Netherlands. [53]Instituto René Rachou, fundação Oswaldo Cruz, Belo Horizonte, Minas Gerais, Brazil. [54]Center for Clinical Research and Prevention, Bispebjerg and Frederiksberg Hospital, København, Denmark. [55]Department of

Clinical Medicine, Faculty of Health and Medical Sciences, University of Copenhagen, Copenhagen, Denmark. [56]Institute of Health and Wellbeing, School of Health and Wellbeing, College of Medical, Veterinary and Life Sciences, University of Glasgow, Glasgow, UK. [57]Human Genetics Center, Department of Epidemiology, Human Genetics, and Environmental Sciences, School of Public Health, The University of Texas Health Science Center at Houston, Houston, TX, USA. [58]Centre for Cognitive Ageing and Cognitive Epidemiology, University of Edinburgh, Edinburgh, UK. [59]Department of Neurology, University of Lübeck, Lübeck, Germany. [60]Fred Hutchinson Cancer Center, University of Washington, Seattle, WA, USA. [61]Labormedizinisches zentrum Dr. Risch, Vaduz, Liechtenstein. [62]Faculty of Medical Sciences, Private University in the Principality of Liechtenstein, Triesen, Liechtenstein. [63]Center of Laboratory Medicine, University Institute of Clinical Chemistry, University of Bern, Inselspital, Bern, Switzerland. [64]Department of Medical Epidemiology and Biostatistics, Karolinska Institutet, Stockholm, Sweden. [65]Greater Bay Area Institute of Precision Medicine (Guangzhou), Fudan University, Nansha District, Guangzhou, China. [66]Epidemiological Cardiology Research Center (EPICARE), Wake Forest School of Medicine, Winston Salem, NC, USA. [67]Dept. of Biochemistry, Cardiovascular Research Institute Maastricht (CARIM), Maastricht University, Maastricht, the Netherlands. [68]Institute of Human Genetics, Genetic Epidemiology, University of Muenster, Muenster, Germany. [69]NIHR Barts Cardiovascular Biomedical Research Centre, Barts and The London School of Medicine and Dentistry, Queen Mary University of London, London, UK. [70]Department of Internal Medicine, Erasmus Medical Center, Rotterdam, the Netherlands. [71]Department of Medicine, School of Medicine, University of North Carolina, Chapel Hill, NC 27599, USA. [72]Department of Medical, Surgery and Health Sciences, University of Trieste, Trieste, Italy. [73]MRC Human Genetics Unit, Institute of Genetics and Cancer, University of Edinburgh, Western General Hospital, Edinburgh, UK. [74]Netherlands Heart Institute, Utrecht, the Netherlands. [75]Durrer Center for Cardiovascular Research, Amsterdam, the Netherlands. [76]Department of Clinical Epidemiology, Leiden University Medical Center, Leiden, the Netherlands, Leiden, the Netherlands. [77]Department of Public Health and Primary Care, Leiden University Medical Center, Leiden, the Netherlands, Leiden, the Netherlands. [78]Department of Biomedical Sciences, University of Copenhagen, Copenhagen, Denmark. [79]Institute of Cardiovascular and Medical Sciences, University of Glasgow, Glasgow, UK. [80]Department of Health Systems and Population Health, University of Washington, Seattte, WA, USA. [81]Department of Internal Medicine, Faculdade de Medicina, Universidade Federal de Minas Gerais, Brazil, Belo Horizonte, Minas Gerais, Brazil. [82]Cardiology Service and Telehealth Center, Hospital das Clínicas, Universidade Federal de Minas Gerais, Belo Horizonte, Brazil, Belo Horizonte, Minas Gerais, Brazil. [83]Departments of Pediatrics and Human Genetics, David Geffen School of Medicine at UCLA, Los Angeles, CA, USA. [84]Department of Cardiology, Heart and Lung Division, University Medical Center Utrecht, Utrecht, the Netherlands. [85]Nuffield Department of Population Health, University of Oxford, Oxford, UK. [86]Department of Epidemiology, Erasmus MC University Medical Center, Rotterdam, the Netherlands. [87]Department of Physiology and Biophysics, University of Mississippi Medical Center, Jackson, MS, USA. [88]Department of Cardiology, Beth Israel Deaconess Medical Center, Boston, MA, USA. [89]APHP, Cardiology Department, Pitié-Salpêtrière Hospital, Paris 75013, France. [90]APHP, Département de Génétique, Centre de Référence Maladies Cardiaques Héréditaires, Pitié-Salpêtrière Hospital, Paris 75013, France. [91]Cardiovascular Health Research Unit, Division of Cardiology, Department of Medicine, University of Washington, Seattle, WA, USA. [92]Genomics and Translational Biomedicine, College of Health and Life Sciences, Hamad Bin Khalifa University, Doha, Qatar. [93]Department of Quantitative Health Sciences, Lerner Research Institute, Cleveland Clinic, Cleveland, OH, USA. [94]Department of Medicine, Cardiovascular Division, Johns Hopkins University, School of Medicine, Baltimore, MD, USA. [95]These authors jointly supervised this work: Larisa G. Tereshchenko, Patricia B. Munroe. ✉e-mail: tereshl@ccf.org; p.b.munroe@qmul.ac.uk

