## [Peer Review File · Nature Communications]

Genetic architecture of spatial electrical biomarkers for cardiac arrhythmia and relationship with cardiovascular diseaseREVIEWER COMMENTS

Reviewer #1 (Remarks to the Author):

To the editors:

The manuscript "Genetic architecture of spatial electrical biomarkers for cardiac arrhythmia and relationship with cardiovascular disease" describes a genetic study of this new cardiac function biomarker in up to 160,000 people. This multi-ancestry meta-analysis generates a wealth of data; I am pleased to see that all summary statistics will be shared with the GWAS catalog.

The authors performed numerous standard post-GWAS analyses which help to identify the likely causal genes, relevant pathways and processes and relationships with related cardiac traits and cardiovascular diseases.

Overall, I think this paper has lots of good substrate. I would like to see the description of the biology developed a bit more.

First, as spQRSTa and fQRSTa are relatively unknown measures, I do think a couple additional sentences would be warranted. What do these measures mean with regards to the structure of the heart? What is a "normal" value and why?

Second, I think there should be a clearer separation between the "results" where several methods are used to identify potential causal genes, and the "discussion" where the possible relevance of these causal genes to cardiac function is discussed.

For example, ADPRHL1 and MYH7 are identified on page 8 (variant-level annotation) and then a biological rationale is provided on page 8. Similarly, FAM135B is mentioned on page 11 (Hispanic/Latino and African-ancestry specific findings) with the biological rationale given there too.

I really like Figure 6 and I think this should be the centerpiece of the discussion (if not the whole paper). I think it would be appropriate to add many other likely causal genes, some mentioned in the text and some only discussed in Sup Tab 14: RYR2, NACA, CDH13, PITX2, CASQ2, DES, SCN5A. Many of these are well known from prior GWAS on related traits but may be unknown to readers new to the area. These genes should also be mentioned in the text, if only to say something like: "we replicated previously identified loci at RYR2, PITX2, ...".

I think this Figure 6 also makes clear that the candidate gene prioritization scheme (header of Sup Tab 14) is probably inappropriate. The genes presented in Figure 6 and in the Discussion are consistent primarily with being the closest genes at their respective loci, having relevant OMIM and mouse knock-out phenotypes and being selected by DEPICT. eQTL coloc, HiC and PrediXcan seem to find evidence for "unlikely causal genes" as often as they do for the "likely causal genes". I am not sure how the prioritization scheme was defined, but it seems inconsistent with the data presented in Sup Tab 14.

One additional change I'd suggest for Figure 6 is to flag that RYR2 and ACTN2 are implicated by a single variant. While there are certainly cases where a single variant impacting a single trait through multiple transcripts, this is less common, and it is not obvious to a casual reader that this is the assumption going in to Figure 6.

Other scientific points:

1) I think the conclusion that "there is limited support for the involvement of genes encoding cardiac ion channels" needs to be better qualified. What about SCN5A?

2) I think the section heading "Relationship between spQRSTa and cardiomyopathies" is misleading since the MR result did not find support for a causal relationship. Maybe a heading like "No evidence for a causal relationship between spQRSTa and cardiomyopathies" would be more appropriate

One grammatical point made the paper almost unreadable for me. Specifically, when the word "however" is used to join to independent clauses it should have a semicolon before it and a comma afterwards. This comes up on almost every page, starting with the abstract:

"They are independent risk predictors for arrhythmia, however the underlying biology is largely unknown."

Should be:

"They are independent risk predictors for arrhythmia; however, the underlying biology is largely unknown."

Finally, I don't find that the current Figure 4 adds much to the paper. It takes a long time to deduce the exact overlap between any pair of traits.

Other minor points.

What does a bold row in Supp Table 9. The legend says both:

rows in bold are where this condition is met.

Rows in bold represent previously unreported loci.

What does a bold "candidate gene" mean in Supp Table 14.

Reviewed by Eric Fauman, Ph.D.

Executive Director, Pfizer Worldwide Research & Development

Reviewer #2 (Remarks to the Author):

Young and colleagues investigated genetic architecture of spatial electrical biomarkers for cardiac arrhythmia (spQRSTa and fQRSTa) by GWAS and its relationship with cardiovascular disease by phenome-wide scanning, identifying 7 loci that have not been reported for any other ECG measures plus potential biology reflected by these derived measures. The authors performed GWAS meta-analysis and associated in silico analyses extensively and their results show some novel findings of interest. However, there are several points, which the authors need to consider.

1. In this study, the authors focused on spatial electrical biomarkers (which are derived from vectorcardiogram) as quantitative traits for GWAS. From the clinical point of view, the usefulness of the vectorcardiogram was well documented in late 1900's but the vectorcardiogram should no longer be considered a routine cardiac test and should be requested only for a specific clinical purpose (Surawicz et al. JACC 1986) apart from several recent general population studies, e.g., GEHCO (ref.8). Given these circumstances, the rationale for focusing on two biomarkers (spQRSTa and fQRSTa) derived from the vectorcardiogram in this study needs to be more clearly described. In the previous GWAS meta-analysis (ref. 13), global electrical heterogeneity (GEH) traits, including SAI QRST and SVG magnitude as well as spQRSTa, were used as a phenotype. Nevertheless, SAI QRST and SVG magnitude are not used but fQRSTa is included instead in the current GWAS meta-analysis. Please explain more details about the criteria for selecting intermediate phenotype traits in such a way.
2. The previous GWAS meta-analysis (ref. 13) identified 10 loci that showed genome-wide significant association with GEH traits in white or joint ancestry. However, 7 (of 10) loci did not appear to be replicated in the current, larger-scale GWAS meta-analysis; this is either because spQRSTa by itself was not significantly associated at these 7 loci in the previous study or because the corresponding associations could not be validated in the current larger study. Although the sample size was relatively

modest ($N=13,826$), the previous study was consisted of the ARIC Study and CHS; both are part of the current study ($N\sim 159,715$). Therefore, if the failure to replicate these 7 loci in the current study is due to differences in statistical power and/or ethnic diversity (i.e., data for 3 ethnic groups were combined in the current study), the authors should be more careful about the robustness of genetic associations with spQRSTa (in particular, at 7 loci that have not been reported for any other ECG measures) even in the current GWAS meta-analysis scaled over 150K individuals.

In this context, please demonstrate the association results (Tables S4 and S5) separately by ethnic group and I^2 statistics/P-values for ethnic heterogeneity. In Table S3, a large part of European-descent individuals appears to be derived from UK Biobank, whereas 10,769 Europeans are shown to be included in the ARIC Study plus CHS. Thus, it may be feasible to look at the reproducibility between the two groups (UK Biobank vs the ARIC Study plus CHS (i.e., ref.13 sample)) and/or between UK Biobank and the other cohorts as a post-hoc replication analysis.

3. In the similar vein, the advantage of using spatial electrical biomarkers for GWAS needs to be discussed more objectively and described more explicitly. That is, in terms of identifying ECG trait-associated loci, the number of novel loci ($N=7$) is relatively small and a large part of the loci overlaps with those that have been already reported for the other ECG traits. Moreover, statistical significance of association for 3 of 7 novel loci is relatively modest, at the level of $P=1\sim 5E-8$, in consideration of the lack of replication stage in the current GWAS. Again, it is preferable to replicate the association signals at least for 7 novel loci before proceeding with detailed discussion about candidate genes at the individual loci.

Hence, the advantage of using spatial electrical biomarkers for GWAS is rather expected to provide opportunities for studying cardiac biology and disease beyond conventional ECG measures, about which the readers as well as the current reviewer will expect to know further, in addition to the inter-trait overlap of GWAS hits as shown in Fig. 4.

4. More details about vector-cardiology transformations need to be described, since the current GWAS meta-analysis seems to have used measures derived from digitalized ECG data via transformation of the 12-lead ECG. In this respect, the reviewer does not see the reason why the numbers of samples considerably differ between analyses of spQRSTa and fQRSTa, if the identical 12-lead ECG data could be used for transformation. Please explain. Also, are there any participating cohorts that have used not the trait data derived from 12-lead ECG but those measured independent of 12-lead ECG?

5. By comparison of the multi-ancestry GWAS hits between spQRSTa (Table S4) and fQRSTa (Table S5), only 3 (out of 11) fQRSTa hits was found to overlap between the two traits, although genetic correlation between the two traits is shown to be high ($r_g=0.61$) according to Figure S6. Looking at Q-Q plots (Figure S2), this is likely due to the stronger effect size for each variant detectable in GWAS of spQRSTa than fQRSTa at the genome-wide level.

Reviewer #3 (Remarks to the Author):

The study undertakes large-scale GWAS of spatial (spQRSTa) and frontal (fQRSTa) QRS-T angles, ECG measures that are risk factors for arrhythmias, and identifies numerous significant loci, some of which are unique to these ECG measures. This is potentially a valuable dataset offering insights into the underlying biology of these traits.

A major focus of the study (and key element of all GWAS) is the annotation of identified loci to highlight and prioritise the likely causal genes. Given the conclusions drawn about the likely underlying biology of these traits, this is an important step but is not performed or presented in a particularly clear manner. There are a number of sections of the Results describing some of the approaches (e.g. CADD scores, eQTLs). The key dataset here seems to be Table S14, which summarises the evidence for loci-gene associations across a number of different methods and describes the role of the putative causal genes. Strangely, this key table gets just a single mention in the manuscript.

For this locus annotation in Table S14, the authors describe 11 methods which are apparently weighted from 1 to 11. However, no justification is given for these weightings or the order of the

evidence classes. Indeed, some seem at odds with recent efforts to define loci-gene prioritisation approaches - e.g. the OpenTargets machine learning approach identified distance features as the most informative metric (PMID:34711957), and yet "nearest gene" seems to be the least prioritised in this study. Similarly, "relevant Mendelian disease gene" is likely to be highly informative for mapping loci in traits like these.

I appreciate the substantial efforts that seem to have gone into these analyses and in the production of this table. However, I think a clearer and more evidence-based approach would benefit this study. Could the authors incorporate metrics from the OpenTargets L2G pipeline and then additionally include trait-specific evidence lines to complement this? I also find this table quite difficult to parse, particularly for the multi-gene loci. Perhaps sticking to one gene per row (with potentially multiple rows for each locus)?

Minor points:

The high genetic correlation between the traits is interesting to note, as is the lower heritability estimates for fQRSTa. But given that all of the significant lead variants for fQRSTa mapped within spQRSTa significant loci, I'm not sure of the value of the section on follow-up of fQRSTa loci?

"26 (42.6%), 27 (44.3%) and 26 (42.6%) lead variants for PR, QRS and HR, respectively, mapped to multi-ancestry spQRSTa loci" - don't these percentages refer to the spQRSTa loci? The phrasing makes it sound like 42.6% of PR lead variants map to the spQRSTa loci.

In the Discussion, TAOK2 is described twice in the same paragraph.

NCOMMS-22-20949-T

Genetic architecture of spatial electrical biomarkers for cardiac arrhythmia and relationship with cardiovascular disease

We thank the reviewers for their comments and suggestions. We believe the revised manuscript has been strengthened, adding clarity to our findings. We have responded in full to each comment and highlighted subsequent revisions to the manuscript in yellow. We have also indicated the location in the manuscript where the corresponding tracked changes can be identified.

REVIEWER COMMENTS

Reviewer #1 (Remarks to the Author):

The manuscript “Genetic architecture of spatial electrical biomarkers for cardiac arrhythmia and relationship with cardiovascular disease” describes a genetic study of this new cardiac function biomarker in up to 160,000 people. This multi-ancestry meta-analysis generates a wealth of data; I am pleased to see that all summary statistics will be shared with the GWAS catalog. The authors performed numerous standard post-GWAS analyses which help to identify the likely causal genes, relevant pathways and processes and relationships with related cardiac traits and cardiovascular diseases. Overall, I think this paper has lots of good substrate. I would like to see the description of the biology developed a bit more.

1) As spQRSTa and fQRSTa are relatively unknown measures, I do think a couple additional sentences would be warranted. What do these measures mean with regards to the structure of the heart? What is a “normal” value and why?

Thank you for your comment. We appreciate your suggestion and agree that these are interesting points. These ECG traits are markers of subclinical cardiac structure and function abnormalities and previous studies have reported a low to moderate association in regression analyses with MRI-derived indices (r^2 0.1 for LV function, 0.2 for end-diastolic function) (Biering-Sorensen et al, PMID: 29496680). Other publications evaluating the relationship of ECG-derived markers of left ventricular hypertrophy with measurements from imaging data, suggest these two modalities may capture distinct information (Bacharova et al, PMID: 25542394). Normal values for the spQRSTa and fQRSTa vary according to the population studied and any underlying disease processes. For example, a spQRSTa $>130^\circ$ in men and $>116^\circ$ in women was associated with sudden cardiac death in dialysis patients from the hospitals of Leiden and Amsterdam (de Bie MK et al, PMID 23024335). Aro et al (PMID: 22183749) used a different cut off ($>100^\circ$) in their Finnish cohort of middle-aged individuals from the general population. Therefore, while a wider QRS-T angle is associated with increased cardiovascular morbidity and mortality, however precise cut offs have not yet been defined (Oehler et al, PMID: 25201032).

Therefore, as these ECG measures are related to cardiac structure but may capture distinct information, and normal values are yet to be defined, we have focused text in the introduction section on their electrophysiological properties.

Introduction P6:

“These markers include the spatial (spQRSTa) and frontal (fQRSTa) QRS-T angles, which are the angles between the directions of ventricular depolarization and repolarization in 3- and 2-dimensional space, respectively (Fig. 1)³. Previous experimental and theoretical studies have shown that a wider QRS-T angle is determined through local variation in action potential duration and morphology^{4,5}.”

2) Second, I think there should be a clearer separation between the “results” where several methods are used to identify potential causal genes, and the “discussion” where the possible relevance of these causal genes to cardiac function is discussed.

For example, ADPRHL1 and MYH7 are identified on page 8 (variant-level annotation) and then a biological rationale is provided on page 8. Similarly, FAM135B is mentioned on page 11 (Hispanic/Latino and African-ancestry specific findings) with the biological rationale given there too.

Thank you for this suggestion to clarify results reporting and strengthen the biological interpretation in the discussion section. We have moved any biological rationale of findings from the results to the discussion.

Sentences removed from the results section:

P8: *MYH7* encodes a beta-myosin heavy chain sarcomeric protein and is associated with the development of inherited cardiomyopathies¹⁷.

P9: The function of *ADPRHL1* in humans has yet to be established, however RNA expression is enriched in heart, skeletal and tongue muscle^{19,20}. In *Xenopus laevis*, *ADPRHL1* regulates protein function through post-transcriptional modification, and knock-out of this gene causes loss of myofibril assembly in ventricular cardiomyocytes²¹.

P11-12: *VAV2* is a guanine nucleotide exchange factor for Ras-related GTPases and modulates receptor-mediated angiogenic responses^{29,30}. Knockout mice exhibit signs of left ventricular hypertrophy, cardiac fibrosis and hypertension³¹.

P12: Knockdown of *FAM135B* in iPSC lines reduces spinal motor neuron survival and contributes to neurite defects as seen in spinal and bulbar muscular atrophy, which is associated with cardiac arrhythmia and structural abnormalities³²⁻³⁴.

P12-13: Little is known about *CCDC60*, however RNA expression is enhanced in smooth muscle, brain, fallopian tube and testis tissues^{19,20}, and it is located within a genomic region associated with neuropsychiatric disorders and learning disability^{35,36}.

Sentences moved and integrated into the discussion:

P17: "A substantial proportion of lead candidate genes at spQRSTa loci are associated with development of inherited cardiomyopathies in humans (including *MYH7*, *TTN*, *TNNT2*, *MYBPC3*, *DSP*, *RBM20*; Fig.6)³⁴. There was also support for genes with non-Mendelian roles in cardiac myogenesis, including *ADPRHL1*, *NACA* and *NFIA*. The function of *ADPRHL1* in humans has yet to be established; however, knockout of *ADPRHL1* in *Xenopus laevis* causes loss of myofibril assembly in ventricular cardiomyocytes and prevents ventricular outgrowth³⁵."

P18: Multiple findings support a role for angiogenesis and arterial development in modulating the spQRSTa, including candidate genes (*ALDH1A2*, *ANGPT1*, and *VAV2*), significant enrichment of GO-terms (coronary vascular development and vasculogenesis), and associations identified in PheWAS (arterial embolism, thrombosis and hypertension). *VAV2*, a candidate gene identified in Hispanic/Latino ancestry-specific analyses, is a guanine nucleotide exchange factor for Ras-related GTPases and modulates receptor-mediated angiogenic responses^{40,41}. Knockout mice for this gene exhibit signs of left ventricular hypertrophy, cardiac fibrosis and hypertension⁴²."

P19: "However, we identified an unreported locus in African ancestry-specific analyses (candidate gene *FAM135B*). Knockdown of *FAM135B* in iPSC lines reduces spinal motor neuron survival and contributes to neurite defects as seen in spinal and bulbar muscular atrophy. These disorders are associated with cardiac arrhythmia and structural abnormalities⁵⁴⁻⁵⁶."

3) I really like Figure 6 and I think this should be the centerpiece of the discussion (if not the whole paper). I think it would be appropriate to add many other likely causal genes, some mentioned in the text and some only discussed in Sup Tab 14: RYR2, NACA, CDH13, PITX2, CASQ2, DES, SCN5A. Many of these are well known from prior GWAS on related traits but may be unknown to readers new to the area. These genes should also be mentioned in the text, if

only to say something like: “we replicated previously identified loci at RYR2, PITX2, ...”.

Thank you for your comments on Figure 6. We have modified the figure to include additional candidate genes. Now 35/61 loci are represented in the figure. We have included a column in Supplementary Table 14 to indicate whether a candidate gene from a locus is included in the figure. To improve visual interpretation, we have removed numbering to indicate the bioinformatic evidence supporting each gene and referred the reader to Supplementary Table 14 instead.

We have also now included text to highlight well known genes from prior GWAS in related traits.

Discussion P18:

“Previous theoretical studies suggested that the spQRSTa reflects abnormalities of ventricular repolarization due to abnormal depolarization³. We identified shared genetic influences and loci overlapping with mainly PR interval and QRS duration. We also report loci that are shared across multiple ECG traits including *NFIA*, *CASQ2*, *RYR2*, *TTN*, *SCN5A*, *PITX2*, *CDKN1A*, *PLN*, *NACA* and *NDRG4* (Fig. 6, Supplementary Table 15).”

Amended Figure 6 (P44):

Figure 6: Illustration of candidate genes at spQRSTa multi-ancestry loci and their potential function

Candidate genes are grouped according to potential roles in embryonic development, cardiac structure and function. *RYR2* and *ACTN2* are candidate genes from the same locus. A summary of the bioinformatic evidence for each gene is presented in Supplementary Table 14. Created using BioRender.com

4) I think this Figure 6 also makes clear that the candidate gene prioritization scheme (header of Sup Tab 14) is probably inappropriate. The genes presented in Figure 6 and in the Discussion are consistent primarily with being the closest genes at their respective loci, having relevant OMIM and mouse knock-out phenotypes and being selected by DEPICT. eQTL coloc, HiC and PrediXcan seem to find evidence for “unlikely causal genes” as often as they do for the “likely causal genes”. I am not sure how the prioritization scheme was defined, but it seems inconsistent with the data presented in Sup Tab 14.

Thank you for raising this. A similar comment was also made by reviewer 3. Our approach to candidate gene prioritisation was to summarise evidence from all the different bioinformatic analyses we did and a literature review of the genes at a locus. We then indicate as a candidate gene the gene at the locus with the most support. For some loci we indicate a second gene as well if there is support from multiple analyses.

As suggested by reviewer 3, we have now also input lead variants into the Open Targets “locus to gene pipeline”. This tool used a machine learning model to learn weights for each evidence source and generate a “locus to gene score” to prioritize genes at a given locus. It is not possible to run this pipeline directly on our summary data as indicated by the original developers (<https://github.com/opentargets/genetics-l2g-scoring>). However, information on variants from the summary statistics of other datasets analysed by the developers of the pipeline is publicly available. At 11 loci there is no data available for a lead variant (or proxy [$r^2 > 0.8$]) in our study. In addition, the pipeline does not consider trait-specific information. Therefore, we have used this data to supplement the evidence generated by the bioinformatic analyses in this study, along with trait-relevant gene information from OMIM, knockout mouse phenotypes and literature review. At 35/50 loci (70%) where data from the pipeline was available, the same candidate gene was prioritized indicating good agreement. At 15 loci where a different gene was suggested by the pipeline, trait-specific evidence (such as mouse knock-out phenotypes or DEPICT gene prioritization) supported an alternative candidate gene, or the “gene to locus” score was low for multiple top genes indicating low confidence in prioritizing a single gene.

We have modified the structure of Supplementary Table 14 to make it easier for the reader to review evidence for each candidate gene and compare with other genes at the locus. We have also made the following amendments to the manuscript to clarify candidate gene reporting and prioritization.

Results P11:

“Candidate Gene prioritization

A summary of bioinformatic annotations for all spQRSTa multi-ancestry loci is provided in Supplementary Table 14. These findings have been supplemented with additional trait-relevant information from: Online Mendelian Inheritance in Man (OMIM)²⁴; and International Mouse Phenotyping Consortium²⁵ (IMP); the Human Protein Atlas²⁶; and PubMed literature reviews for each candidate gene. We also performed lookups of each lead variant in the Open Targets Genetics ‘Locus to Gene’ machine learning gene prioritization pipeline for further annotations (Supplementary Table 14)²⁷.”

Methods P26:

“The output of all bioinformatic analyses were pooled and supplemented with trait relevant information from OMIM²⁴ and IMP (www.mousephenotype.org)²⁵ databases, the Human Protein Atlas²⁶ (proteinatlas.org) and a PubMed literature review (Supplementary Table 14). We also performed a look up of each lead variant in the Open Targets Genetics “Locus to Gene” machine learning pipeline, which uses supervised learning to weight evidence from different sources and prioritize genes at a locus²⁷. This database does not include trait-specific information in the pipeline and therefore it is used to supplement the analyses performed for this work. For each locus, the candidate gene with the most support across all lines of evidence is indicated. We also included a second gene if there is support from multiple analyses”

Supplementary Table 14:

Addition of a gene to locus column and modification of the header to clarify how genes were selected for the “likely candidate gene column”.

“A candidate gene at each locus was identified as the gene with the most support after pooling all lines of evidence from trait-specific bioinformatic findings, trait-relevant findings from the Online Mendelian Inheritance in Man (OMIM) and International Mouse Phenotyping consortium (IMP) databases, expression data from the human protein atlas (proteinatlas.org) literature review and output from the open targets locus to gene pipeline. Gene expression is defined by Protein Atlas as enriched in heart tissue if at least four-fold higher mRNA compared to other tissue and enhanced if at least 4-fold higher mRNA compared to the average in all other tissue. Protein expression is categorized by Protein Atlas according to staining intensity (negative, weak, moderate or strong) and fraction of stained cells (<25%, 25-75% or >75%). eQTL; expression quantitative trait locus, COLOC; Colocalization, PP; Posterior Probability, RAA; Right atrial appendage, LV; Left ventricle, RV; Right ventricle, AA; Aorta artery, CA; Coronary artery, L2G score; Locus to gene score as output by the open targets genetics platform.”

5) One additional change I'd suggest for Figure 6 is to flag that RYR2 and ACTN2 are implicated by a single variant. While there are certainly cases where a single variant impacting a single trait through multiple transcripts, this is less common, and it is not obvious to a casual reader that this is the assumption going in to Figure 6.

Thank you for this comment. We have now mentioned this in the figure legend.

Other scientific points:

1) I think the conclusion that “there is limited support for the involvement of genes encoding cardiac ion channels” needs to be better qualified. What about SCN5A?

We have clarified this statement where we mean to indicate that in comparison to other ECG traits, there are fewer loci with candidate genes encoding cardiac ion channels. We have also modified a subsequent sentence in the discussion to clarify our meaning.

Discussion:

P16:

“Among spQRSTa and fQRSTa loci, there are fewer genes for cardiac ion channels, in contrast to findings for other ECG traits.”

P18:

“In comparison to previously reported loci for QT and JT, there is less support for the involvement of genes encoding cardiac potassium channels, which are important determinants of ventricular repolarization and common targets of existing anti-arrhythmics⁴⁷.”

2) I think the section heading “Relationship between spQRSTa and cardiomyopathies” is misleading since the MR result did not find support for a causal relationship. Maybe a heading like “No evidence for a causal relationship between spQRSTa and cardiomyopathies” would be more appropriate

Thank you for this suggestion. We have made the amendment to the section heading.

3) One grammatical point made the paper almost unreadable for me. Specifically, when the word “however” is used to join to independent clauses it should have a semicolon before it and a comma afterwards. This comes up on almost every page, starting with the abstract: “They are independent risk predictors for arrhythmia, however the underlying biology is largely unknown.”. Should be: “They are independent risk predictors for arrhythmia; however, the underlying biology is largely unknown.”

Thank you for highlighting this. We have corrected this grammatical error at all points in the manuscript and also reviewed wording throughout the manuscript to improve the communication of our findings.

4) Finally, I don’t find that the current Figure 4 adds much to the paper. It takes a long time to deduce the exact overlap between any pair of traits.

Thank you for this comment. We have revised the figure to show the names of loci rather than a number indicating how many loci overlap. We hope this modification improves interpretation of the figure and makes it more interesting for the reader to study.

Figure 4: Overlap of multi-ancestry spQRSTa loci with ECG measures

Venn diagram showing spQRSTa multi-ancestry loci where a lead variant reported for another ECG measure maps within the locus boundaries. For this figure, ECG measures shown are PR interval (cardiac conduction), QRS duration (ventricular depolarization), QT and JT intervals (ventricular repolarization) and heart rate (HR). Overlap was declared if a lead variant for these ECG measures mapped to within $\pm 500\text{kb}$ or $r^2 > 0.1$ of a lead variant at a spQRSTa locus. Some loci overlap with other ECG traits (not visualised here but presented in Supplementary Table 15). At seven spQRSTa loci, no overlap was observed with any ECG trait (blue circle bottom right).

Other minor points.

- 1) What does a bold row in Supp Table 9. The legend says both: rows in bold are where this condition is met. Rows in bold represent previously unreported loci.**

Thank you for identifying this typo. It should read only “rows in bold are where this condition is met”. We have now corrected the header.

2) What does a bold "candidate gene" mean in Supp Table 14.

We have removed this in the new Supplementary Table

Reviewer #2 (Remarks to the Author):

Young and colleagues investigated genetic architecture of spatial electrical biomarkers for cardiac arrhythmia (spQRSTa and fQRSTa) by GWAS and its relationship with cardiovascular disease by phenome-wide scanning, identifying 7 loci that have not been reported for any other ECG measures plus potential biology reflected by these derived measures. The authors performed GWAS meta-analysis and associated in silico analyses extensively and their results show some novel findings of interest. However, there are several points, which the authors need to consider.

1. In this study, the authors focused on spatial electrical biomarkers (which are derived from vectorcardiogram) as quantitative traits for GWAS. From the clinical point of view, the usefulness of the vectorcardiogram was well documented in late 1900's but the vectorcardiogram should no longer be considered a routine cardiac test and should be requested only for a specific clinical purpose (Surawicz et al. JACC 1986) apart from several recent general population studies, e.g., GEHCO (ref.8). Given these circumstances, the rationale for focusing on two biomarkers (spQRSTa and fQRSTa) derived from the vectorcardiogram in this study needs to be more clearly described. In the previous GWAS meta-analysis (ref. 13), global electrical heterogeneity (GEH) traits, including SAI QRST and SVG magnitude as well as spQRSTa, were used as a phenotype. Nevertheless, SAI QRST and SVG magnitude are not used but fQRSTa is included instead in the current GWAS meta-analysis. Please explain more details about the criteria for selecting intermediate phenotype traits in such a way.

We agree that vectorcardiographic markers are currently not used in routine clinical practice. The manuscript by Surawicz in 1986 (JACC) looked specifically at the role of electrocardiographic measures for the detection of ventricular chamber dilatation and hypertrophy in comparison with echocardiography. However, over the last decade there has been a resurgence of interest in these markers for risk prediction of other conditions that has also coincided with computational advances that make their calculation more feasible in the clinical setting. Furthermore, as you suggested in your second comment, studying vectorcardiographic derived measures in a genome-wide association study has potential to identify biology that is not captured by classical ECG measures such as PR interval, QRS duration and the QT interval. Our study is a large multicentre collaborative study and therefore it was not feasible in this current work to evaluate all markers that are included under the "global electrical heterogeneity" umbrella, such as SAI QRST and SVG magnitude. We decided to focus on vectorcardiographic measures that have been most extensively evaluated in the literature for our first study.

We have amended the introduction to provide greater justification for the study of these measures:

Introduction P6:

"While vectorcardiographic measures are not currently used in routine clinical practice, there has been a resurgence of interest in their potential clinical utility, which has coincided with computational advances for efficient calculation of these markers. Recent studies have reported associations of the spQRSTa and fQRSTa with risk for arrhythmogenesis, sudden cardiac death and cardiac-related mortality⁶⁻⁸. In a population-based study, an abnormal spQRSTa was associated with a five-fold increased risk of cardiac and sudden death. No other conventional

cardiovascular or ECG measure provided higher hazard ratios⁹. These measures may also be broad markers of cardiovascular risk, and associations have been reported with cardiomyopathies and cardioembolic stroke^{10,11}. Improved knowledge of these markers will increase our understanding of these clinical relationships and has potential to identify new biology that is not captured by conventional ECG measures.”

2.

A) The previous GWAS meta-analysis (ref. 13) identified 10 loci that showed genome-wide significant association with GEH traits in white or joint ancestry. However, 7 (of 10) loci did not appear to be replicated in the current, larger-scale GWAS meta-analysis; this is either because spQRSTa by itself was not significantly associated at these 7 loci in the previous study or because the corresponding associations could not be validated in the current larger study. Although the sample size was relatively modest (N=13,826), the previous study was consisted of the ARIC Study and CHS; both are part of the current study (N=~159,715). Therefore, if the failure to replicate these 7 loci in the current study is due to differences in statistical power and/or ethnic diversity (i.e., data for 3 ethnic groups were combined in the current study), the authors should be more careful about the robustness of genetic associations with spQRSTa (in particular, at 7 loci that have not been reported for any other ECG measures) even in the current GWAS meta-analysis scaled over 150K individuals.

B) In this context, please demonstrate the association results (Tables S4 and S5) separately by ethnic group and I^2 statistics/P-values for ethnic heterogeneity. In Table S3, a large part of European-descent individuals appears to be derived from UK Biobank, whereas 10,769 Europeans are shown to be included in the ARIC Study plus CHS. Thus, it may be feasible to look at the reproducibility between the two groups (UK Biobank vs the ARIC Study plus CHS (i.e., ref.13 sample)) and/or between UK Biobank and the other cohorts as a post-hoc replication analysis.

C) In the similar vein, the advantage of using spatial electrical biomarkers for GWAS needs to be discussed more objectively and described more explicitly. That is, in terms of identifying ECG trait-associated loci, the number of novel loci (N=7) is relatively small and a large part of the loci overlaps with those that have been already reported for the other ECG traits. Moreover, statistical significance of association for 3 of 7 novel loci is relatively modest, at the level of $P=1\sim 5E-8$, in consideration of the lack of replication stage in the current GWAS. Again, it is preferable to replicate the association signals at least for 7 novel loci before proceeding with detailed discussion about candidate genes at the individual loci. Hence, the advantage of using spatial electrical biomarkers for GWAS is rather expected to provide opportunities for studying cardiac biology and disease beyond conventional ECG measures, about which the readers as well as the current reviewer will expect to know further, in addition to the inter-trait overlap of GWAS hits as shown in Fig. 4.

The only GWAS meta-analysis to date for GEH traits was by Tereshchenko et al 2018, they identified three independent genome-wide significant loci for the spQRSTa (*HAND1*, *TBX3* and *NFIA*). These loci were also reported for other GEH measures (*HAND1*: SVG elevation, SVG magnitude; *HAND1*: SVG azimuth; *NFIA*: SVG azimuth). These three loci were replicated in our study and were the strongest associations (by *P*-value). The other 7 genome-wide significant loci reported in the original study, were for other phenotypes – SAI QRST and SVG magnitude, and were not genome-wide significant for spQRSTa in the original study. While all these vectorcardiographic measures are conceptually connected underneath the same umbrella term of “Global Electrical Heterogeneity”, they are different phenotypes and share low correlations (0.243 and -0.249 for spQRSTa v SAI QRST and SVG magnitude, respectively). Therefore, there is not an expectation that previously reported loci for SAI QRST and SVG should also be genome-wide significant for spQRSTa or fQRSTa. Despite this however, 3 lead variants reported for SAI QRST also map within the boundaries ($\pm 500\text{kb}$ or $r^2 > 0.1$) of

spQRSTa loci in our multi-ancestry meta-analysis (*SCN5A*, *MYBPC3*, *NDRG4*). We have now reported this overlap in the manuscript.

We have performed the largest discovery analysis for spQRSTa and fQRSTa to date. It is therefore not possible to validate findings in this study due to a lack of a suitably sized replication dataset. However not withstanding validation being required, we believe it is of value to highlight our observations of the 7 loci not previously reported for other ECG measures. We have modified the text to indicate validation is required and reduced some of the detailed discussion of these loci.

Results P13:

“Despite the low genetic correlations observed genome-wide, 26 (42.6%), 27 (44.3%) and 26 (42.6%) lead multi-ancestry spQRSTa variants mapped to previously reported PR, QRS and HR loci, respectively (Supplementary Table 15). Fewer variants mapped to reported QT and JT loci (19 [31.1%] and 14 [23%], respectively) (Fig. 4). Of the 7 loci reported for the global electrical heterogeneity trait SAI QRST, 3 lead variants mapped within the boundaries of spQRSTa loci (*SCN5A*, *MYBPC3* and *NDRG4*).”

Discussion P18-19:

“There was no overlap with other ECG traits at 7 multi-ancestry spQRSTa loci. Candidate genes at these loci include: *AHNAK2*, which encodes a large nucleoprotein that localises to the Z-band region of mouse cardiomyocytes and may have a role in excitation-contraction coupling through effects on L-type voltage-gated calcium channels; *SGCG*, a component of the subsarcolemmal cytoskeleton; and *ALDH1A2*, which encodes an enzyme responsible for early embryonic retinoic acid synthesis, a process that is critical for normal cardiac and arterial development⁴⁸⁻⁵¹. Another candidate gene *TAOK2*, is a protein kinase most studied for its role in dendritic spine maturation⁵². More recently, *TAOK2* has been identified in tethering the endoplasmic reticulum to microtubules. We report another locus, *MACF1*, that is also involved in microtubule organization^{53,54}. Validation of these loci is required.”

Supplementary Tables 4 and 5:

We have updated Supplementary Tables 4 and 5 as suggested. For each lead variant in the multi-ancestry, European, Hispanic/Latino and African ancestry meta-analyses, we have added corresponding association results for each ancestral group (effect allele frequency, beta, standard error and *P*-value), along with *I*² statistics and *P*-values for ethnic heterogeneity.

3. More details about vector-cardiology transformations need to be described, since the current GWAS meta-analysis seems to have used measures derived from digitalized ECG data via transformation of the 12-lead ECG. In this respect, the reviewer does not see the reason why the numbers of samples considerably differ between analyses of spQRSTa and fQRSTa, if the identical 12-lead ECG data could be used for transformation. Please explain. Also, are there any participating cohorts that have used not the trait data derived from 12-lead ECG but those measured independent of 12-lead ECG?

All ECG data used in this study was calculated from the resting 12-lead ECG. On recruitment to our study, some cohorts did not still have access to the digitized ECG signal but had QRS and T-wave axes available for calculation of the fQRSTa. These studies therefore contributed to the fQRSTa meta-analysis but not the spQRSTa meta-analysis. We have expanded the text in the methods on calculation of these measures.

Methods P21:

“A GWAS was performed by each participating cohort for the spQRSTa (mean) and fQRSTa. If the spQRSTa was not already calculated and digitized ECGs were available, it was derived by transformation of the 12-lead ECG using previously published algorithms⁵⁷. In brief,

after applying a bandpass butterworth filter and signal averaging to reduce noise, orthogonal X, Y and Z vector beats were estimated using Kors' regression matrix⁶³. The spQRSTa was subsequently calculated as the angle between mean QRS and T-wave spatial vector loops^{57,58}. The fQRSTa was defined as the absolute difference between QRS and T-wave frontal plane axes ($fQRSTa = abs[QRS-axis - T-axis]$)³. Values for both phenotypes are between 0 and 180°."

4. By comparison of the multi-ancestry GWAS hits between spQRSTa (Table S4) and fQRSTa (Table S5), only 3 (out of 11) fQRSTa hits was found to overlap between the two traits, although genetic correlation between the two traits is shown to be high (rg=0.61) according to Figure S6. Looking at Q-Q plots (Figure S2), this is likely due to the stronger effect size for each variant detectable in GWAS of spQRSTa than fQRSTa at the genome-wide level.

All multi-ancestry fQRSTa loci overlapped with spQRSTa loci. At three loci (*SCN5A*, *RBM20* and *TBX3*), the same lead variant is reported for both traits. For the remaining 8 loci, the lead variant for fQRSTa mapped within the boundaries of a multi-ancestry spQRSTa locus (definition of overlap $\pm 500\text{kb}$ or $r^2 > 0.1$, whichever was greater). Of these, all fQRSTa lead variants were within 86kb of a spQRSTa lead variant, except rs1724411 (17:43669931:C:T), which was $> 500\text{kb}$ away from spQRSTa lead variant rs2668692 (17:44293020:A:G, candidate gene *KANSL1*). However these two variants had an r^2 of 0.64 and are therefore within the same locus.

Reviewer #3 (Remarks to the Author):

The study undertakes large-scale GWAS of spatial (spQRSTa) and frontal (fQRSTa) QRS-T angles, ECG measures that are risk factors for arrhythmias, and identifies numerous significant loci, some of which are unique to these ECG measures. This is potentially a valuable dataset offering insights into the underlying biology of these traits.

A major focus of the study (and key element of all GWAS) is the annotation of identified loci to highlight and prioritise the likely causal genes. Given the conclusions drawn about the likely underlying biology of these traits, this is an important step but is not performed or presented in a particularly clear manner. There are a number of sections of the Results describing some of the approaches (e.g. CADD scores, eQTLs). The key dataset here seems to be Table S14, which summarises the evidence for loci-gene associations across a number of different methods and describes the role of the putative causal genes. Strangely, this key table gets just a single mention in the manuscript.

For this locus annotation in Table S14, the authors describe 11 methods which are apparently weighted from 1 to 11. However, no justification is given for these weightings or the order of the evidence classes. Indeed, some seem at odds with recent efforts to define loci-gene prioritisation approaches - e.g. the OpenTargets machine learning approach identified distance features as the most informative metric (PMID:34711957), and yet "nearest gene" seems to be the least prioritised in this study. Similarly, "relevant Mendelian disease gene" is likely to be highly informative for mapping loci in traits like these.

I appreciate the substantial efforts that seem to have gone into these analyses and in the production of this table. However, I think a clearer and more evidence-based approach would benefit this study. Could the authors incorporate metrics from the OpenTargets L2G pipeline and then additionally include trait-specific evidence lines to complement this? I also find this table quite difficult to parse, particularly for the multi-gene loci. Perhaps sticking to one gene per row (with potentially multiple rows for each locus)?

Thank you for your comment, which was also raised by Reviewer 1. We appreciate that the numbering system was confusing to the reader and did not clearly describe how we prioritized a candidate gene at a locus. We identified the most likely candidate gene by reviewing all lines of evidence and selecting the gene with the most support. We have also included a second gene when there is support from multiple analyses. We have clarified this in the manuscript text in the relevant sections (please see response to Reviewer 1).

Thank you for suggesting the Open Targets L2G pipeline. It is not possible to run this pipeline directly on our summary data as indicated by the original developers (<https://github.com/opentargets/genetics-l2g-scoring>). However, information on variants from the summary statistics of other datasets analysed by the developers of the pipeline are publicly available. Using this data for loci where the variant (or proxy, $r^2 > 0.8$) had been analysed (50 loci), at 35 (70%) loci Open Targets prioritized the same gene that we selected indicating good agreement. For the remaining 15 loci, trait-specific findings suggested another candidate gene, or the locus to gene score was low for multiple genes indicating the pipeline has low confidence with prioritizing a single gene at the locus. As explained in the response to Reviewer 1, we have now included this data as supporting evidence in Supplementary Table 14.

We have also modified Supplementary Table 14 as you have suggested, to display the literature review for each gene in separate rows. Along with rearranging columns to clearly separate bioinformatic analyses and database look ups of trait-relevant findings for each trait, we believe the table is now more accessible to the reader. As described in the response to Reviewer 1, we have also modified the relevant sections in the results, discussion and Supplementary Table 14.

Minor points:

The high genetic correlation between the traits is interesting to note, as is the lower heritability estimates for fQRSTa. But given that all of the significant lead variants for fQRSTa mapped within spQRSTa significant loci, I'm not sure of the value of the section on follow-up of fQRSTa loci?

Thank you for your comment. As all lead variants for fQRSTa map within spQRSTa loci, we agree with the reviewer and have reduced the text in this section. We have focused on highlighting additional information from the bioinformatic analyses that was not reported for spQRSTa loci.

Results P12-13:

“Follow-up of loci for the frontal QRS-T angle

Three variants at two loci were significant eQTL variants (LV [*SSXP10*, *RP11-632C17_A.1*], coronary artery [*GNAZ1*]), but there was no support for colocalization (Supplementary Table 9). Eight genes were significant in the TWAS, and overlapped with spQRSTa genes, except for two (*CEP85L* and *MMP11*) (Supplementary Table 10). Tissue-specific promoter interactions were identified for variants at two loci that were not reported for spQRSTa loci (lead variant rs10885011; *FAM124A* and *DLEU7*, rs5030613; *BCR*) (Supplementary Table 11a and b). An unreported locus identified in the African ancestry fQRSTa meta-analysis was not GWS in spQRSTa analyses. The gene nearest to the lead signal is *CCDC60* (Coiled-Coil Domain Containing 60).”

"26 (42.6%), 27 (44.3%) and 26 (42.6%) lead variants for PR, QRS and HR, respectively, mapped to multi-ancestry spQRSTa loci" - don't these percentages refer to the spQRSTa loci? The phrasing makes it sound like 42.6% of PR lead variants map to the spQRSTa loci.

Thank you for identifying this issue with the phrasing. The percentages do refer to the spQRSTa loci and we have now corrected the sentence.

Results P13:

“Despite the low genetic correlations observed genome-wide, 26 (42.6%), 27 (44.3%) and 26 (42.6%) lead multi-ancestry spQRSTa variants mapped to reported PR, QRS and HR loci, respectively (Supplementary Table 15). Fewer variants mapped to reported QT and JT loci (19 [31.1%] and 14 [23%], respectively) (Fig. 4).”

In the Discussion, TAOK2 is described twice in the same paragraph.

Thank you for highlighting this. We have removed the duplicated sentence.

REVIEWERS' COMMENTS

Reviewer #1 (Remarks to the Author):

Thank you for the detailed responses to my concerns. I think the figure 6 is fabulous and the overall paper is much stronger now.

Reviewer #2 (Remarks to the Author):

The authors have revised their manuscript satisfactorily in response to the reviewer's comments. There are no other comments.

Reviewer #3 (Remarks to the Author):

I thank the authors for their thorough responses to the reviewer comments which have improved the manuscript and congratulate them for a very interesting and impactful study.

NCOMMS-22-20949-A

Genetic architecture of spatial electrical biomarkers for cardiac arrhythmia and relationship with cardiovascular disease

We thank the reviewers for their comments and valuable feedback which has strengthened the manuscript. We have also completed all editorial requests and formatting requirements needed for this revision and supply comments in the authors checklist accompanying this submission.